# Cryo-EM structure-based selection of computed ligand poses enables design of MTA-synergic PRMT5 inhibitors of better potency

Wei Zhou[1,2,8], Gaya P. Yadav [3,4,7,8], Xiaozhi Yang[1,2], Feng Qin[5], Chenglong Li [1,2,6✉] & Qiu-Xing Jiang [1,3,4,5✉]

Projected potential of 2.5–4.0 Å cryo-EM structures for structure-based drug design is not well realized yet. Here we show that a 3.1 Å structure of PRMT5 is suitable for selecting computed poses of a chemical inhibitor and its analogs for enhanced potency. PRMT5, an oncogenic target for various cancer types, has many inhibitors manifesting little cooperativity with MTA, a co-factor analog accumulated in MTAP−/− cells. To achieve MTA-synergic inhibition, a pharmacophore from virtual screen leads to a specific inhibitor (11-2 F). Cryo-EM structures of 11-2 F / MTA-bound human PRMT5/MEP50 complex and its *apo* form resolved at 3.1 and 3.2 Å respectively show that 11-2 F in the catalytic pocket shifts the cofactor-binding pocket away by ~2.0 Å, contributing to positive cooperativity. Computational analysis predicts subtype specificity of 11-2 F among PRMTs. Structural analysis of ligands in the binding pockets is performed to compare poses of 11-2 F and its redesigned analogs and identifies three new analogs predicted to have significantly better potency. One of them, after synthesis, is ~4 fold more efficient in inhibiting PRMT5 catalysis than 11-2 F, with strong MTA-synergy. These data suggest the feasibility of employing near-atomic resolution cryo-EM structures and computational analysis of ligand poses for small molecule therapeutics.

[1] Department of Medicinal Chemistry, College of Pharmacy, University of Florida, Gainesville, FL 32610, USA. [2] Department of Biochemistry and Molecular Biology, College of Medicine, University of Florida, Gainesville, FL 32610, USA. [3] Department of Microbiology and Cell Science, University of Florida, Gainesville, FL 32611, USA. [4] Laboratory of Molecular Physiology and Biophysics, Hauptman-Woodward Medical Research Institute, Buffalo, NY 14203, USA. [5] Department of Physiology and Biophysics, the State University of New York at Buffalo, Buffalo, NY 14214, USA. [6] Center for Natural Products, Drug Discovery and Development, University of Florida, Gainesville, FL 32610, USA. [7] Present address: G.P.Y at the Department of Biochemistry and Biophysics, Texas A &M University, College Station, TX 77843, USA. [8] These authors contributed equally: Wei Zhou, Gaya P. Yadav. ✉email: lic@ufl.edu; qxjiang@ufl.edu

The so-called "resolution revolution" in single particle cryo-EM has produced hundreds of structures of 2.5–4.5 Å resolutions in ten years[1–8]. Many of these structures represent biomacromolecules complexed with small molecule agonists, antagonists, or modulators[2,9]. Although single particle cryo-EM often lags behind crystallography in resolution because a majority of cryo-EM structures are still of 3.5 Å resolution or worse, it is still expected to usher a new era in structure-based drug design (SBDD), which has been successful with high-resolution crystal structures and fragment-based design. Although atomic resolutions (~1.2 Å) for high-symmetry complexes were achieved recently[10,11], most near-atomic resolution cryo-EM structures of low- or no-symmetry have been resolved at 2.5–4.5 Å. These developments promised a high potential of cryo-EM SBDD, which may be realized for many important biological or pharmacological targets; but such potential is still not well materialized, in part because most cryo-EM structures (>85%) are not resolved to 1.0–2.5 Å, where X-ray SBDD has been quite successful[2]. With advancements in computational analysis of ligand configurations (poses)[12,13], virtual screening, docking and energy minimization of ligands could be utilized to enhance cryo-EM SBDD. We started to test this possibility in 2015, using human protein arginine methyltransferase 5 (PRMT5) as a target.

PRMT5 is epigenetically important[14–16], because it regulates gene transcription[17], cell signaling[18], RNA splicing[19], DNA repair[20], chromatin remodeling[21] and cell cycle[22]. It is overexpressed in various human cancers[23–28], and its inhibition causes cell death[29,30], especially in cancer cells lacking 5′-methylthioadenosine phosphorylase (MTAP). ~15% of all human cancers lack MTAP (MTAP-) and accumulate by 5–20 folds a metabolic derivative—5′-methylthioadenosine (MTA)—of S-adenosyl-L-methionine (SAM), the cofactor for PRMT5 catalytic activity. High MTA sensitizes MTAP−/− cells to partial PRMT5 inhibition that spares MTAP+ cells[31–33]. PRMT5 is therefore a compelling anti-cancer target[34,35].

A type II PRMT, PRMT5, catalyzes symmetric arginine dimethylation of histone proteins (e.g., H4R3 and H3R8)[36] and non-histone proteins, such as p53 and NF-κB[37,38]. It consumes SAM to form an ω-N$^G$-monomethyl arginine (MMA), releasing an S-adenosyl-L-homocysteine (SAH) as a byproduct. A second SAM molecule is used to generate ω-N$^G$,N′$^G$-symmetric dimethylarginine (SDMA) as the final product[39–41]. PRMT5 activity requires a crucial partner, called methylosome-associated protein 50 (MEP50)[36,42], which dramatically augments PRMT5 catalysis by increasing substrate recognition and presentation[43,44]. X-ray structures of PRMT5 from *C. elegans*, *X. laevis* and *H. sapiens*[39,43,45] revealed a conserved triosephosphate isomerase (TIM) barrel at the N-terminal half, a Rossmann-fold domain for cofactor binding and a β-sandwich domain between the two for substrate binding and dimerization. MEP50, a 7-WD40 β-propeller protein, binds PRMT5 via contacts with the TIM barrel, making a hetero-octameric PRMT5:MEP50 complex one of an apparent D2 symmetry[39]. The octamer methylates a broad spectrum of substrates, including COPR5[46], SWI/SNF[47], pICln[48], Riok1[49], Menin[50], etc., and making targeted PRMT5 inhibition a potential cancer therapy.

Currently, all PRMT5 inhibitors fall into four classes based on their modes of action. The first class is cofactor-site competitive inhibitors[51], which are SAM analogs with ribose and adenine moieties strongly favored by the Rossmann-fold domain. The second class is substrate-site competitive inhibitors with high potency and selectivity[52]. The third class is allosteric modulators indirectly altering the canonical binding sites[53]. The fourth class includes inhibitors that suppress binding of substrate adapter proteins (SAPs) or decrease PRMT5:MEP50 stability[54,55]. Many inhibitors have been discovered and designed[56]. Two molecules, JNJ-64619178 of the first class and GSK-3326595 of the second

class, entered clinic trials for multiple cancer types in 2018[51]. They appear less potent in preclinical studies than recently-identified class 2 molecules, MRTX1719 and its analogs (like MRTX9768), from high-throughput fragment screen and X-ray SBDD[57,58]. Moreover, a nucleoside-based inhibitor targeting a conserved cysteine residue (C449) inside the SAM-binding site[59] lacks specificity among SAM-binding enzymes. On the other hand, most class 2 inhibitors rely on SAM or its analogs for PRMT5 inhibition[60], but are less potent in the presence of MTA[58,61]. For *MTAP−/−* cancer cells, class 2 inhibitors working in synergy with MTA are desired to suppress or kill them more specifically while sparing *MTAP+* cells.

Available structural data are insufficient for understanding interactions between a Class 2 inhibitor and MTA. A crystal structure (PDB: 3UA4) of *apo C. elegans* PRMT5:MEP50 complex differs significantly from that of the liganded human complex[45], whereas the *apo* human complex has been resistant to crystallization. The first structure of the human complex with a SAM analog A9145C and a histone H4 substrate peptide (PDB: 4GQB) was a milestone and triggered competition in X-ray SBDD[39,58]. However, all published crystal structures of human PRMT5 were obtained in complex with one or more ligands[37] and reveal little on their low synergy with MTA.

Search for MTA-synergic PMRT5 inhibitors that work more potently in MTAP−/− cancer cells asks for a different strategy. Although chemical screen against PRMT5/MTA complexes may provide a practical, yet costly, approach (as Mirati Therapeutics did)[57,58], we asked whether combination of near-atomic resolution cryo-EM structures with computational analysis of compound poses constitutes a different, but much less expensive, approach for selecting the most probable pose and using it to design higher-potency inhibitors. The first cryo-EM structure of PRMT5:MEP50 complexed with a cofactor analog (dehydrosinefungin) was solved at a 3.7 Å resolution[62], which would be insufficient for our purpose. We reasoned that the high-quality phase information in cryo-EM structures of ~3.0 Å resolutions may provide sufficient constraints for large ligands and allow proper docking and selection among possible poses optimized by computational analysis[2,9]. The top candidate(s) will be accurate enough for structure-based redesign and reselection, ultimately leading to more potent inhibitors. In this paper, we tested this strategy on a PRMT5 inhibitor initially discovered by virtual screen[63]. We determined its stably-bound pose, unraveled the chemical basis for its synergy with MTA, designed and reselected derivatized compounds, and synthesized one of the designed compounds to confirm its higher potency in enzyme inhibition. Our results suggest that after more tests this non-crystallographic strategy probably has broader applications to cryo-EM-based molecular therapeutics.

## Results

**Virtual screening led to chemically different PRMT5 inhibitors.** Pharmacophores differing from EPZ015666 in the Cambridge database were screened against PRMT5, yielding a pharmacophore with comparatively better binding efficiency[29]. The resulted pharmacophore was modified manually based on the crystal structure of the catalytic pocket to increase its non-covalent interactions with the residues lining the binding pocket and was synthesized in the Li lab (Fig. 1a). Manual optimization of the molecules benefited from structure-activity relationship (SAR) and docking of the resulting compound into the catalytic pocket. Compounds with a better binding network were synthesized and tested for in vitro inhibition of enzyme activity. Recombinant PRMT5:MEP50 was purified to biochemical homogeneity from *sf*9 cells (Supplementary Fig. 1). Inhibition of

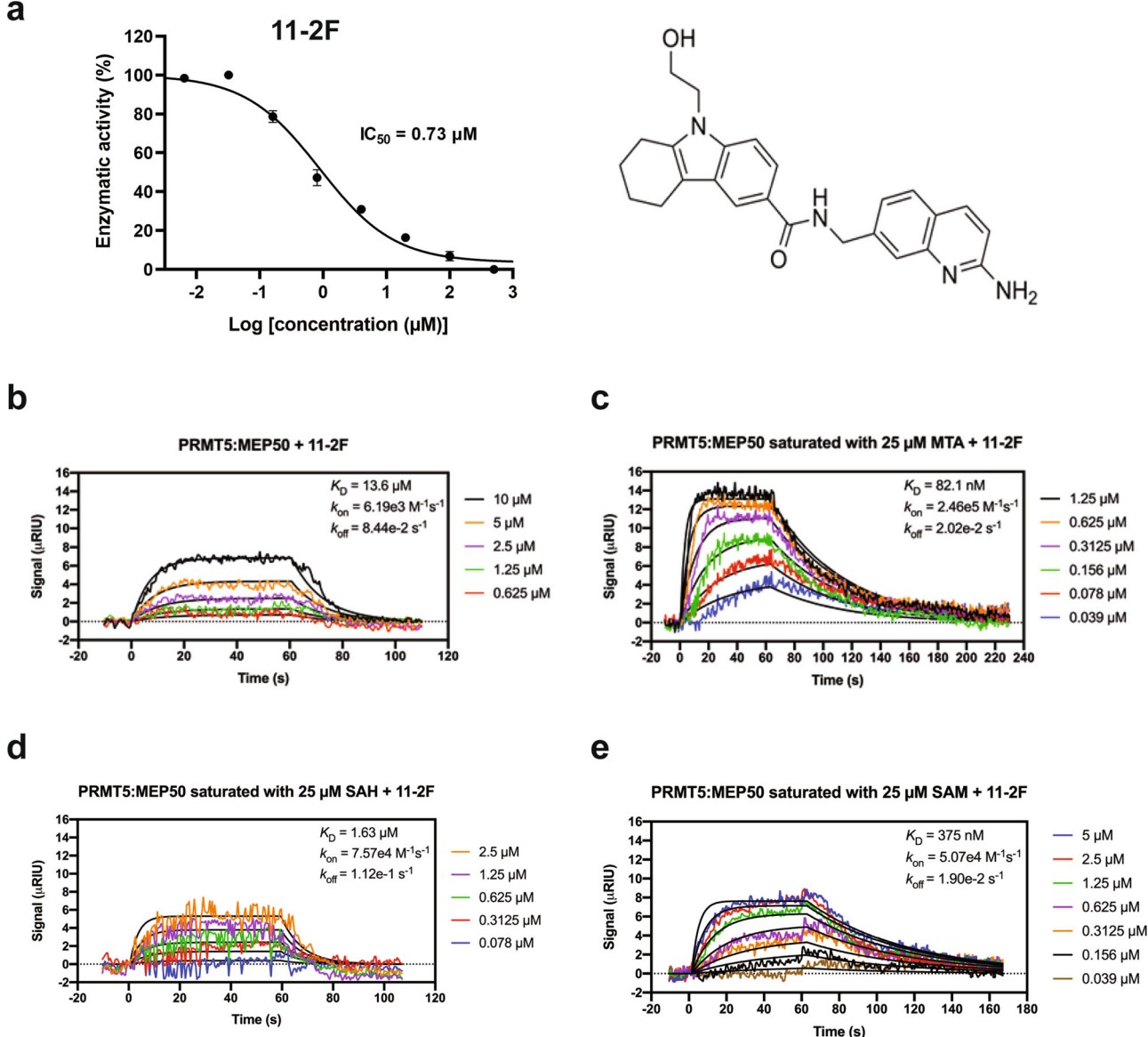

**Fig. 1 An inhibitor 11-2F of PRMT5 exhibits positive cooperativity with MTA. a** Dose-dependent inhibition of enzyme activity by 11-2F. IC50 ~730 nM. The chemical structure of 11-2F is showed on the right. Errors: s.d., $n = 3$. **b** SPR of 11-2F binding and unbinding to PRMT5:MEP50 in the absence of MTA, leading to a calculated $K_D$ ~13.6 μM. **c** SPR of 11-2F interaction with the enzyme in the presence of MTA. $K_D$ ~82 nM. The apparent positive coupling coefficient between 11-2F and MTA is ~166. **d**, **e** SPR of 11-2F binding to PRMT5:MEP50 complex in the presence of SAH (**d**) and SAM (**e**), showing much weaker affinity, 1.6 and 0.38 μM, respectively, and thus much weaker synergy than MTA.

PRMT5 activity (methylation of H4 peptide) was monitored by measuring luminescence from a MTase-Glow detection reagent. One of the compounds, 11-2F, inhibited PRMT5 strongly (IC50: 0.73 ± 0.2 μM). Surface plasma resonance (SPR) studies detected that 11-2F binds to PRMT5:MEP50 with an apparent $K_D$ ~13.6 μM without MTA, but its affinity increased drastically to ~82 nM in the presence of 25 μM MTA (Fig. 1c), much better than its synergy with SAM and S-adenosyl-L-homocysteine (SAH) (Fig. 1d, e). How to increase further the potency of 11-2F in PRMT5 inhibition is a challenging question. Because the 11-2F-bound PRMT5:MEP50 complex resisted crystallization, it made a good candidate for testing the proposed cryo-EM SBDD strategy by selecting compound poses from computation analysis.

**A 3.1 Å cryo-EM structure of 11-2F-bound human PRMT5:MEP50.** The PRMT5:MEP50/MTA/11-2F complex was prepared by mixing stock solutions of MTA and 11-2F with

0.4 mg/ml PRMT5:MEP50 (~0.09 μM) on ice for 30 min before being applied to glow-discharged QuantiFoil grids. Movies collected in a Titan Krios were processed as outlined in the supplementary information (Supplementary Fig. 2), yielding a cryo-EM reconstruction (Coulombic potential map) at a nominal resolution of ~3.1 Å [Fourier Shell Correlation (FSC) at 0.143] (Fig. 2 and Table 1). The estimated resolution agrees well with revolved structural details (Fig. 2a–d). Local resolutions estimated by ResMap vary in 2.4 – 4.1 Å, slightly overestimated relative to that from FSC (Fig. 2b)[64]. An X-ray structure (PDB: 6CKC) was fitted into the cryo-EM map by real-space refinement and manual adjustment (Fig. 2a). The resulted cryo-EM model showed structural differences from the crystal structure (orange; Fig. 2c). The overall changes in the relative positions of PRMT5 and MEP50 (Fig. 2c) were small (RMSD~1.0 Å) and likely resulted from crystal packing forces. Figure 2d shows clearly-resolved densities corresponding to side chains in two

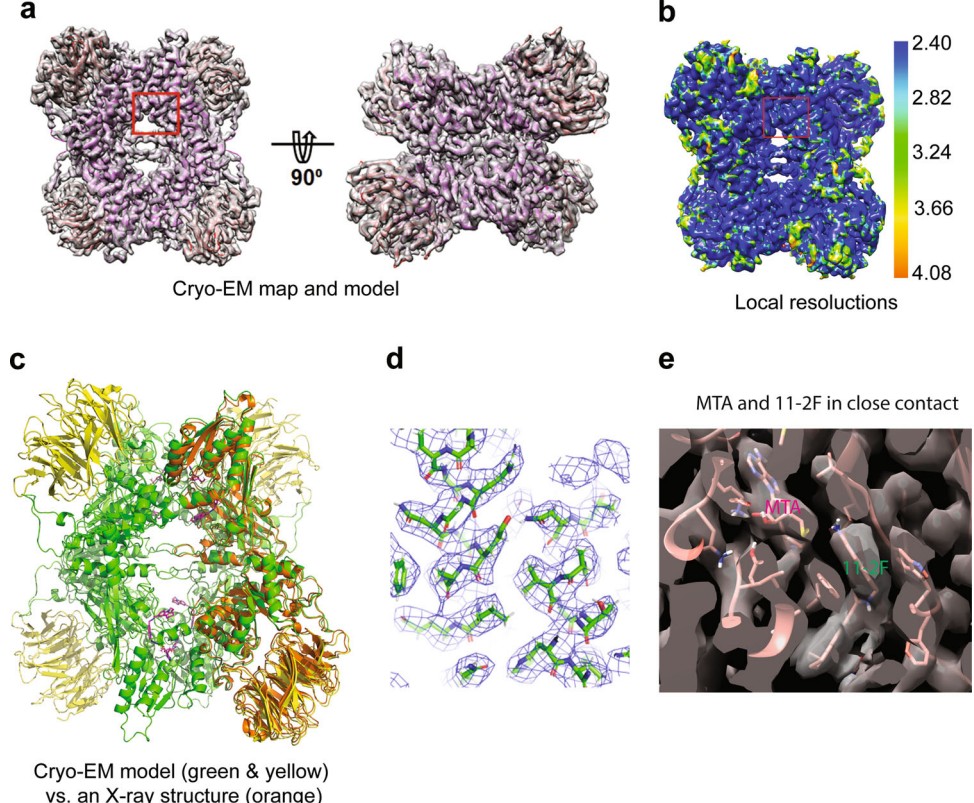

**Fig. 2 Cryo-EM structure of the 11-2F-bound PRMT5:MEP50 complex. a** A 3.1 Å cryo-EM map of the complex in two different orientations with the atomic model built in the density (PRMT5 in pink and MEP50 in red). **b** Local resolution variations of the map estimated by ResMap. **c** Comparison of the cryo-EM model (green PRMT5 and yellow MEP50) with the X-ray structure (orange; PDB: 6CKC). **d** Matching of the side chains in two short helices of the cryo-EM model with the density (blue mesh). **e** Densities for MTA and 11-2F are well defined in the cryo-EM map. The MTA tail is right next to the quinoline part of 11-2F. The models of the protein and the inhibitor are in pink.

neighboring α-helices of PRMT5. More differences occur within the MEP50 domain at the periphery of the complex (Fig. 2c), where all β-sheets were resolved (Fig. 2a). Because MEP50 plays an insignificant role in our structure-based analysis of the compound, its structural differences will not be discussed further.

To our satisfaction, the 3.1 Å cryo-EM map clearly shows the density expected for the inhibitor inside the catalytic site of PRMT5 (Fig. 2e), right next to the co-factor binding site (marked as MTA). At this resolution, the 11-2F density does not distinguish the two-ring quinoline (head part) from the three-ring tetrahydro-carbazole (or cyclo-alkylated indole; tail part). Even though the resolutions in the binding pocket (box in Fig. 2b) (overestimated at 2.6–2.8 Å by ResMap) appear better than the periphery, the two-ring head or the three-ring tail is not resolved clearly. The cryo-EM maps at 2.5–4.0 Å resolutions thus have significant uncertainty for defining accurate poses of low molecular-weight ligands[2]. The fact that the ligand density can be accounted for by its two parts (stick model in Fig. 2e) suggests that the 3.1 Å cryo-EM map probably is suitable for modeling better the compound pose. The density for MTA, is well resolved, and right next to 11-2 F (Figs. 2e and 3a). These structural features in the cryo-EM Coulombic potential map triggered us to test if it is feasible to perform more accurate ligand modeling by molecular docking and energy minimization from non-covalent interactions, pose selection to distinguish the head and tail parts, and structure-based design of 11-2F for further selection.

**Pose selection in the binding sites**. As a positive control for the proposed strategy, we first analyzed the MTA-binding site because a crystal structure is available to check the quality of our

results. We asked whether the cryo-EM density of MTA is sufficiently good to distinguish its right pose among various possible ones from computational analysis. Different MTA poses were generated by software packages for molecular docking and ranked via minimization of binding energy. If the cryo-EM map-based selection is accurate, the resulted pose is expected to be very similar, if not identical, to that determined by X-ray crystallography at a higher reported resolution. 25 million random poses were generated in AUTODOCK to introduce rotational freedom around all rotatable bonds and identify one with the lowest binding energy in each run[12,65,66]. AUTODOCK clustered the poses from 2000 runs internally based on RMSD < 2.0 Å (Supplementary Fig. 3a and Supplementary Table 1). The top pose in the 1st cluster represented ~89% of the 2000 poses from random starting configurations, suggesting that it was heavily favored. The mean binding energy of the first cluster is significantly lower than the second one. Structural comparison found that the top pose of the first cluster differs from the ones from other clusters (Supplementary Fig. S3a). When the top 3 poses from the clustering analysis were compared with the cryo-EM density, the top one (colored green in Fig. 3b) matches better with the cryo-EM density. The top pose overlaps well with the MTA structural model determined by X-ray crystallography (yellow *vs.* cyan, Fig. 3d) and uses the same H-bonds for binding. These results suggested that the AUTODOCK-optimized top poses selected against the cryo-EM density at ~3.1 Å resolution can lead to an accurate binding pose for a compound as small as MTA that contains a two-ring structure of an adenosine.

As more positive controls of our protocol, we generated top poses of SAH (Supplementary Fig. 3b and Supplementary Table 2)

**Table 1 Cryo-EM data collection and modeling statistics.**

| | PRMT5:MEP50:11-2 F | PRMT5:MEP50 |
|---|---|---|
| **Data collection and processing** | | |
| Magnification | 105,000 | 105,000 |
| Voltage (kV) | 300 | 300 |
| Electron exposure (e−/Å$^2$) | 40 | 40 |
| Defocus range (μm) | −0.75 to −3.0 | −0.75 to −3.0 |
| Pixel size (Å) | 0.66 | 0.66 |
| Symmetry imposed | D2 | D2 |
| Initial number of particle images (no.) | 866,240 | 235,210 |
| Final number of particle images (no.) | 207,392 | 101,707 |
| Map sharpening B factor (Å$^2$) | −68 | −79.7 |
| Overall map resolution (Å) | 3.14 | 3.17 |
| FSC threshold | 0.143 | 0.143 |
| **Refinement** | | |
| Initial model used (PDB code) | Apo PRMT map | Generated in Relion3.0 |
| FSC model (0/0.143/0.5) Å | 3.1/3.2/3.4 | 2.9/3.2/3.6 |
| **Model composition (per asymmetric unit)** | | |
| Chains | 2 | 2 |
| Non-hydrogen atoms | 7384 | 7265 |
| Protein residues | 932 | 925 |
| Ligands | 2 | 0 |
| B factors (Å$^2$) (min/max/mean) | 24.99/75.31/47.74 | 61.67/137.13/86.91 |
| **R.M.S. deviations** | | |
| Bond angles (°) | 0.566 | 0.645 |
| Bond distances (Å) | 0.005 | 0.007 |
| **Validation** | | |
| MolProbity score | 0.78 | 1.05 |
| Clash score | 0.69 | 0.89 |
| Poor rotamers (%) | 0.00 | 0.00 |
| **Ramachandran plot** | | |
| Favored (%) | 97.73 | 97.58 |
| Allowed (%) | 2.16 | 1.87 |
| Outliers (%) | 0.11 | 0.55 |
| EMringer Score | 3.02 | 2.84 |

clustering analysis of the docking results found that poses with the head part of 11-2 F deep in the binding pocket (1 and 2 in Supplementary Table 4) are more stable than those with the tail part inserted deeply (4 and 6). The energetic analysis, without details of ordered water molecules, therefore distinguishes the orientations of head and tail parts (Supplementary Figs. 3e, f).

As a control, we tested whether different software packages would generate similar or the same top poses for the same compound against the same structural model. We compared MTA and 11-2F in their respective binding pockets of the cryo-EM structural model using AUTODOCK, AutoDock Vina and SwissDock[65,66,68,69]. The top poses from the three were very similar to each other (Fig. 3i, j), probably because of the same chemical constraints and polarization parameters for respective energy minimization processes. Our data so far showcase that the poses generated from computational analysis can be selected and refined against a ligand density in a cryo-EM map of ~3.1 Å resolution, producing a (most probable) ligand-binding model of high accuracy.

**A cryo-EM Structure of the _apo_ human PRMT5:MEP50 complex.** Comparison of the 11-2F/MTA-bound cryo-EM structure with the known X-ray structures containing other ligands suggests that the flexible loop deduced from the structure of the _C. elegans_ PRMT5 might be induced into an ordered state and make an important part of the compound-binding pocket from the periphery. The flexible loop thus could be used for SBDD. Moreover, a structure of the _apo_ complex would help verify the densities assigned to MTA in the cofactor-binding pocket and 11-2F in the substrate-binding pocket. We obtained a 3.2 Å cryo-EM structure of the _apo_ PRMT5:MEP50 complex (Supplementary Fig. 4 and Fig. 4a). Parameters for data processing and molecular modeling are enumerated in Table 1. The local resolutions of the map vary (Fig. 4b). The structural model fits the density well after real-space refinement (Fig. 4c), except the flexible loop (Fig. 4e). As expected, no densities in the two binding pockets for MTA and 11-2F are visible even at a lower threshold. There is very weak or no density corresponding to most parts of the flexible loop (Fig. 4e), whereas the density of the loop region was fairly strong in the map of the 11-2F/MTA bound complex (yellow, Fig. 4f). In the atomic model for the _apo_ complex, residues 292–294, 304–307, and 312–329 were thus omitted. The modeling of the leftover residues in the loop also harbors higher uncertainty (Table 1).

Since in the _apo_ state, both MTA- and 11-2F-binding pockets are empty, we compared the volume changes of the two pockets between the two cryo-EM structures. The estimated volume of the MTA-binding pocket in the _apo_ state is ~28% smaller than that in the liganded state. The putative substrate-binding pocket in the _apo_ state is larger in volume because the flexible loop is disordered, leaving an open end. Such differences unveil two important points. 1) MTA-binding induces a change in the binding pocket, probably due to induced fit and the ordering of the flexible loop. 2) The substrate-binding pocket in the _apo_ state has a larger volume so that a substrate or an inhibitor (11-2 F) experiences significant freedom in testing different poses before becoming securely bound with the flexible loop making part of the pocket (Fig. 4f).

**Structural basis for synergy between MTA and 11-2F.** Our data in Fig. 1b, c suggest positive cooperativity between MTA and 11-2F, much better than SAM or SAH, probably due to MTA's smaller size. To retain this cooperativity when 11-2F is redesigned, we would like to understand its possible structural underpinnings. Even though the global alignment of the MTA/

and SAM (Supplementary Fig. 3c and Supplementary Table 3) in the known X-ray structural models, and found that the top pose for each agrees well with the respective X-ray model, except minor differences in the flexible tail regions (Fig. 3d, e). These comparisons support our general strategy and argue strongly that the predictions from ligand docking and energy minimization in AUTODOCK are relatively accurate for ring-containing compounds and can be further improved when cryo-EM densities of sufficient resolutions are available to constrain them.

We then applied the same strategy to 11-2F in the substrate-binding site. The top poses from AUTODOCK (Supplementary Figs. 3d, e and Supplementary Table 4) were compared with the cryo-EM map (Fig. 3f), and then refined against the density by a simplified all-atom molecular dynamics calculation in ISODE[67] (Fig. 3g, h). The top pose from AUTODOCK is very close to the final refined model (yellow vs. green, Fig. 3g) with only one rotation of the quinoline group by 180 degrees after refinement, suggesting that a cryo-EM map of 3.1 Å contains sufficient details of the ligand for selecting and refining the top poses generated by computational analysis (Fig. 3h), leading to an accurate model of the ligand, even though individual atoms in two multi-member rings of the quinoline are not resolved in the cryo-EM map. The

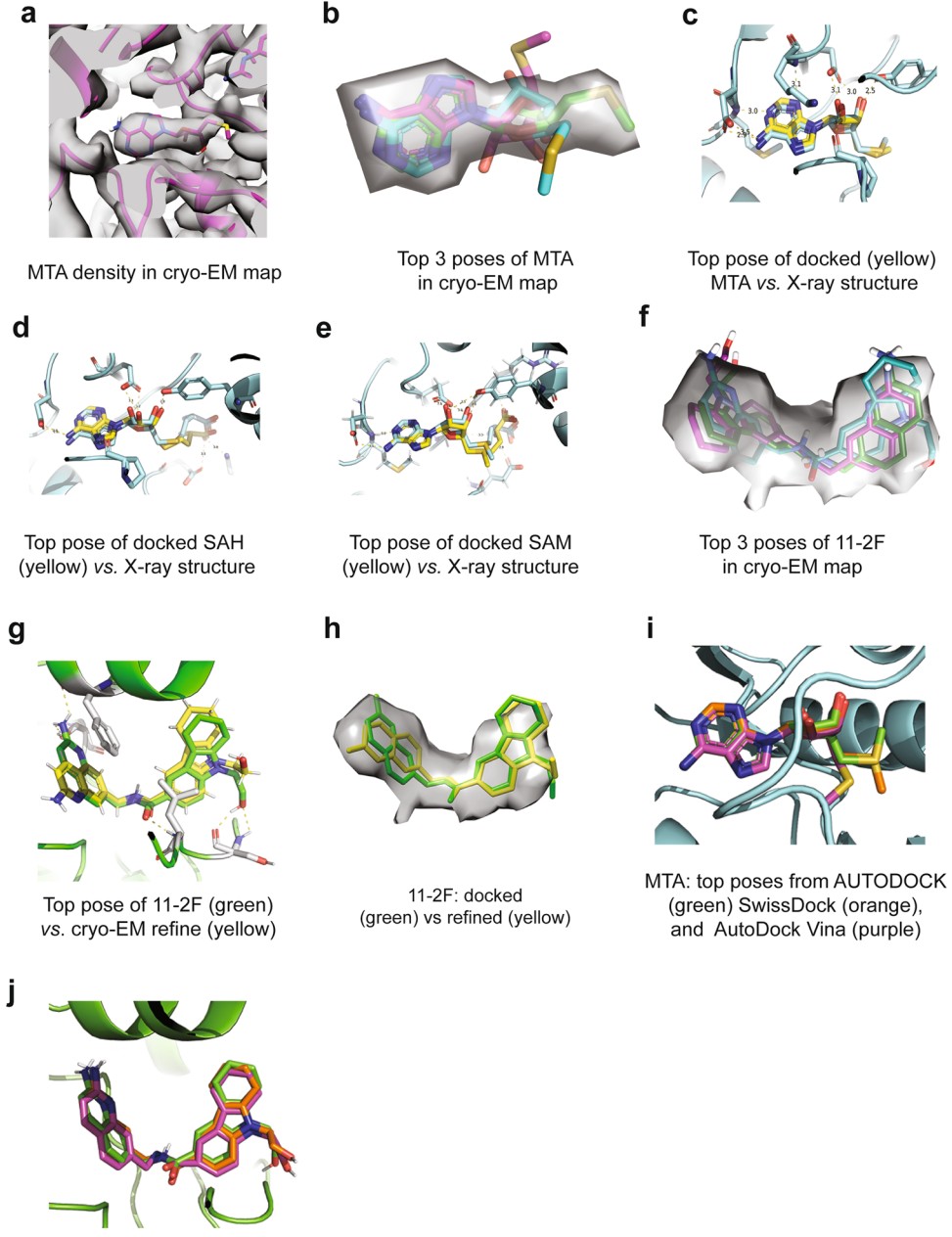

**a** MTA density in cryo-EM map

**b** Top 3 poses of MTA in cryo-EM map

**c** Top pose of docked (yellow) MTA *vs.* X-ray structure

**d** Top pose of docked SAH (yellow) *vs.* X-ray structure

**e** Top pose of docked SAM (yellow) *vs.* X-ray structure

**f** Top 3 poses of 11-2F in cryo-EM map

**g** Top pose of 11-2F (green) *vs.* cryo-EM refine (yellow)

**h** 11-2F: docked (green) vs refined (yellow)

**i** MTA: top poses from AUTODOCK (green) SwissDock (orange), and AutoDock Vina (purple)

**j** 11-2F: top poses from different softwares

**Fig. 3 Cryo-EM density guides selection of top poses of ligands from molecular docking and energy minimization.** Besides the accommodation of most or all atoms of the ligands in their densities, ranking of the ligand-protein interaction energy helps define the most stable poses. **a** Density in the cryo-EM map for MTA. **b** Overlay of top three poses from the AUTODOCK analysis with the cryo-EM density for MTA (gray). The top pose is in green. **c** After refinement, the top pose is almost in exactly the same position and orientation as that from the X-ray structure (cyan). Dashed lines represent the H-bonds in the binding pocket. **d** Top pose of SAH from AUTODOCK agrees with the model from the crystal structure (cyan). **e** Top pose of SAM from AUTODOCK overlaps very closely with the one in the crystal structure (cyan). **f** Top three poses of 11-2F out of AUTODOCK overlap relatively well with the cryo-EM map, which was improved after real-space refinement against the cryo-EM density. **g/h** After optimization in ISOLDE, the top pose (green in **g**) of 11-2F changes slightly with the quinoline ring rotated by 180 degrees around a rotatable bond (in yellow). A simplified all-atomistic MD optimization in ISOLDE enhances the agreement of the model (green) with the ligand density (**h**). The dashed lines in **g** represent key H-bonds for the ligand binding. **i** Comparison of top poses of MTA from three different software packages (AUTODOCK, SwissDock, and AutoDock Vina). **j** Top poses of 11-2F from three packages are nearly identical.

11-2F-bound cryo-EM model and the MTA/H4 peptide-bound X-ray model revealed an RMSD of ~1.0 Å, we looked into the intra-subunit movements at the binding pockets by first aligning the N-terminal TIM barrel domains of the two structures (bottom parts in Fig. 5a) and then comparing relative movement between their catalytic domains. Relative to the X-ray model, the whole cofactor-binding pocket in the cryo-EM structure is shifted upwards by ~2.0 Å at the top, as if 11-2F occupancy of the substrate-binding site pushes the MTA-binding site upward (black arrow in Fig. 5b). Published data showed that MTA and H4 peptide had no positive cooperativity, suggesting that the relative shift of the two bindings sites in the catalytic domain (Fig. 5b) is likely a key contributor to the positive co-operativity between 11-2F and MTA. Such a mechanism could be further

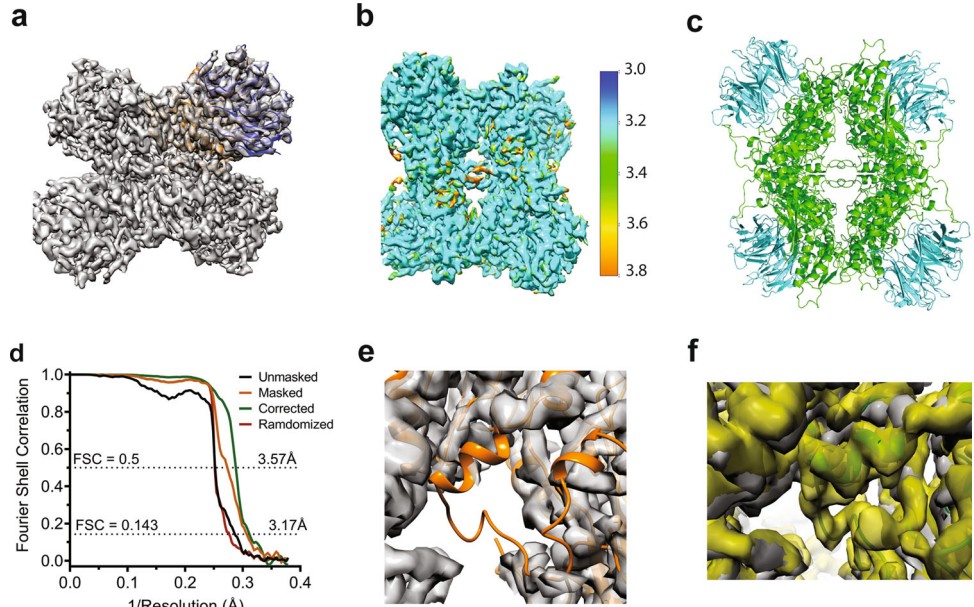

**Fig. 4 Cryo-EM structure of the *apo* human PRMT5:MEP50 depicts the disorder of the flexible loop. a** Cryo-EM map of ~3.2 Å in resolution. **b** Resolution variations estimated by ResMap. **c** The molecular model. **d** FSC for different maps. The corrected map at 3.2 Å was used for modeling. **e** The density map with the model for the ligand-bound complex (orange) highlights the disordered loop region not in the density. **f** The flexible loop contributes to the binding pocket for the ligand at its periphery.

vetted by structures of 11-2F or H4-bound complexes without MTA or in the presence of SAM. Figure 5c highlights the shift of MTA together with the wall of its binding pocket as if the bound 11-2F favors the MTA binding. Such a physical shift retains the key residues that stabilize MTA. Inversely, it is equally probable that the changes induced by MTA-binding favor structural changes of the substrate binding pocket for 11-2F to bind into the groove between the β-barrel domain and the Rossman fold. The mutual interactions therefore result in the positive cooperativity.

This synergistic mechanism for MTA and 11-2F appears different from the positive cooperativity between SAM and the H4 peptide (Fig. 5b) or between SAM and the GSK inhibitor (EPZ015666 in Fig. 5e) because the longer tail of SAM or its analogs (e.g., LLY283 in Fig. 5d) disfavors a similar shift of the cofactor binding pocket. Alternatively, the physical shifts for MTA-binding (Fig. 5c) do not promote the binding of the H4-peptide or the GSK inhibitor so that they have no positive cooperativity with MTA. From this line of thinking, a guiding principle for redesigning 11-2 F would be to preserve interactions between the quinoline ring of 11-2F and residues in the binding pocket, including Glu435, Glu444, Phe327, Trp579, etc., to retain MTA-synergy.

**Predicted subtype specificity of 11-2F among PRMTs**. With the top poses from computational analysis being very close to the final one refined against the cryo-EM density (Fig. 3g), it was tempting to ask whether the same analysis of 11-2F among available structural models of six other PRMT proteins would reveal unique features for 11-2F binding to PRMT5 (Fig. 6). We first identified key residues coordinating 11-2F in the substrate-binding pocket of PRMT5 that are conserved among PRMTs (Supplementary Fig. 5). Carboxyl oxygen of Asp419 shows a strong interaction (O–N = 2.3 Å) with the nitrogen of the amine group in the quinoline ring. The aromatic ring of quinoline is stacked in parallel to sidechains of Phe 327 and Trp579. All residues in the binding pocket, such as Leu312, Thr323, Phe327, Leu336, Gly365, Gly367, Lys393, Glu435, and Ser578, etc.,

participate in coordinating 11-2F by non-covalent interactions (Fig. 6e). The nitrogen atom of the 2-amine group on the quinoline ring and its nitrogen at position 1 form strong H-bonds with catalytic residues Glu 444 & Glu 435 with N-O distances of 2.41 Å and 2.86 Å, respectively. Other residues, including Gln322, Pro311, Lys 333, Leu437, Val503, Leu312, and Ser310, etc., act similarly (Fig. 6e). Comparison of the crystal structure of EPZ015666-bound PRMT5:MEP50 with our 11-2F-bound cryo-EM structure shows that despite small differences in protein-inhibitor interactions for the two inhibitors, PRMT5 takes similar conformations in the presence of the two. This analysis suggests the feasibility to introduce extra interacting groups to the tail part of 11-2F and enhance its potency.

We next used the same docking method to predict the top poses of 11-2F in the binding pockets of PRMT1-4 and 6-7 (Fig. 6a–d, f, g). Sequence alignment of PRMT1-7 in Supplementary Fig. 5 shows that key residues for ligand-protein interactions are conserved in the substate-binding pockets. Based on top poses, the binding pockets in PRMT1, PRMT3, PRMT6, and PRMT7 clearly cannot accommodate the quinoline ring in the right position for the two catalytic Glu residues to interact with it. The PRMT2 is pretty poor because the cyclo-alkylated indole ring of 11-2F has very limited interactions with the binding pocket. The main reason appears that the PRMT2 substrate-binding pocket is too shallow to accommodate the two parts of 11-2F completely, making its binding energy fairly high (Fig. 6h). PRMT4 is probably the only one that might have a relatively good binding affinity because its Glu266 interacts with the 2-amine on the quinoline ring and its Tyr154 hydroxyl interacts with the N-atom inside the quinoline ring. Other interactions next to the cyclo-alkylated indole ring help stabilize the tail part (Fig. 6d). The resulted docking energy in PRMT4 agrees with the predicted interactions in the binding pocket (Fig. 6h). These analyses predict that 11-2F is able to differentiate PRMT5 from other PRMTs due to the chemical differences among their binding pockets. It will be interesting to test if mutations in the binding pockets of PRMT4 can enhance 11-2F binding.

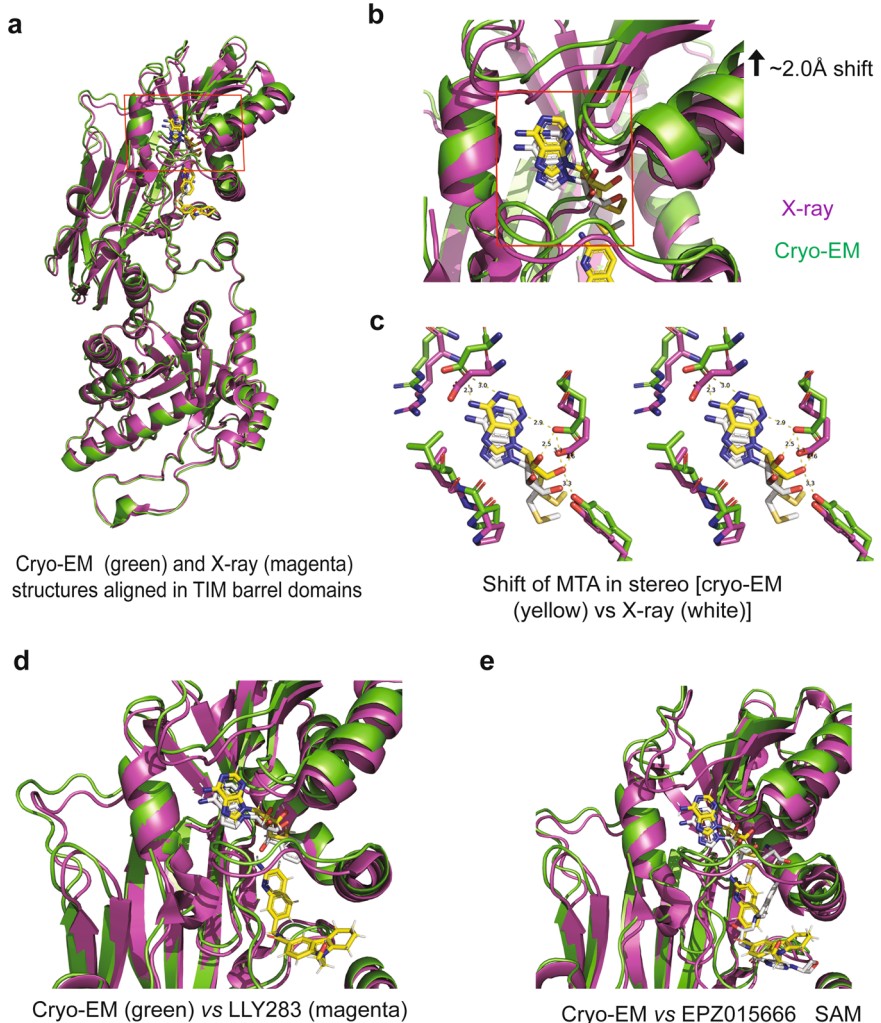

Cryo-EM (green) and X-ray (magenta)
structures aligned in TIM barrel domains

Shift of MTA in stereo [cryo-EM
(yellow) vs X-ray (white)]

Cryo-EM (green) *vs* LLY283 (magenta)

Cryo-EM *vs* EPZ015666 _SAM

**Fig. 5 Structural changes underlying the synergy between 11-2F and MTA. a** After alignment of the TIM domains (bottom) between the cryo-EM model and the crystal structure of the MTA/H4 bound complex (PDB: 5FA5), the ligand-binding domain of the cryo-EM model is shifted upwards, especially more significantly in the MTA-binding pocket (red square zoomed in panel **b**. **b** The red-square in **a** is magnified to highlight the shift (black arrow). The flexible loop region is also shifted upward. The red-square marks the MTA-binding pocket to be showed in panel **c**. **c** A stereo view of the MAT-binding pockets. A push from the 11-2F-binding makes the co-factor binding pocket tighter for MTA binding. **d** The shift of the MTA-binding pocket did not happen in the LLY283-bound structure (magenta; PDB: 6CKC). LLY283 is a SAM-analog. **e** The co-factor binding pocket did not shift in the structure of the EPZ/SAM complex (magenta; PDB: 4 × 61).

**Structure-based design of 11-2F for higher potency in enzyme inhibition**. The above analysis highlighted three principles that should be considered to enhance PRMT5 inhibition by 11-2F analogs. (1) It is important to retain the quinoline ring backbone to maintain the positive cooperativity with MTA. (2) The pi-stacking interactions of Trp579 and Phe327 with the quinoline ring may be enhanced to keep the inhibitor properly oriented, which could be achieved by introducing small groups (-F, or -CH$_3$, or -NH$_3^+$) to the ring. (3) The cyclo-alkylated indole ring of 11-2F is relatively flexible and may be stabilized by introducing H-bonds or electro-static interactions with the binding pocket, especially with the residues on the flexible loop. We used these principles to guide the design of dozens of 11-2F derivatives and utilized computational analysis to predict their most stable poses before ranking them and selecting the most stable ones. Three 11-2F analogs predicted to have higher potency introduce more interactions with the binding pockets (Table 2; Supplementary Figs. 6–8, and Supplementary Tables 5–7). From the constraints in the cryo-EM model (Fig. 6e), the quinazoline parts of these compounds are in almost the same location and orientation as the quinoline of 11-2 F. The predicted

binding affinity is ~18 nM for 11-9 F, and ~1.0 nM for HWIem2104 and 2109, which are in a relative scale (Table 2, right column). In the binding pocket, the 2,4-di-NH$_2$-quinazoline of 11-9 F forms H-bonds with E444, E435, and S439, and its aromatic ring is sandwiched between the sidechains of Phe327 and Trp579. The backbone amino group of F580 also contributes to its binding. When 11-9 F was synthesized and assayed, it inhibits PRMT5 enzyme activity ~4 fold (~180 nM) more potently than 11-2 F does (Figs. 7e vs. 1a), in good agreement with predictions of the docking analysis and energy minimization (Fig. 7d). When a cell-based assay was performed to measure symmetrically dimethylated arginine (SDMA) from PRMT5 activity by western blotting (Fig. 7f; β-actin as loading controls; a full film image is in Supplementary Fig. 9), 11-9F inhibition of SDMA production in the presence of 5.0 μM MTA is ~13.6 fold more potent than without additional MTA (0.0 μM; red vs. black in Fig. 7g). Despite limitations from membrane permeability, 11-9F inhibition shows clear and strong MTA-synergy in cells. Improvement of its membrane permeability or construction of a vehicle to facilitate its membrane crossing will make 11-9F even better in ex vivo or in vivo tests.

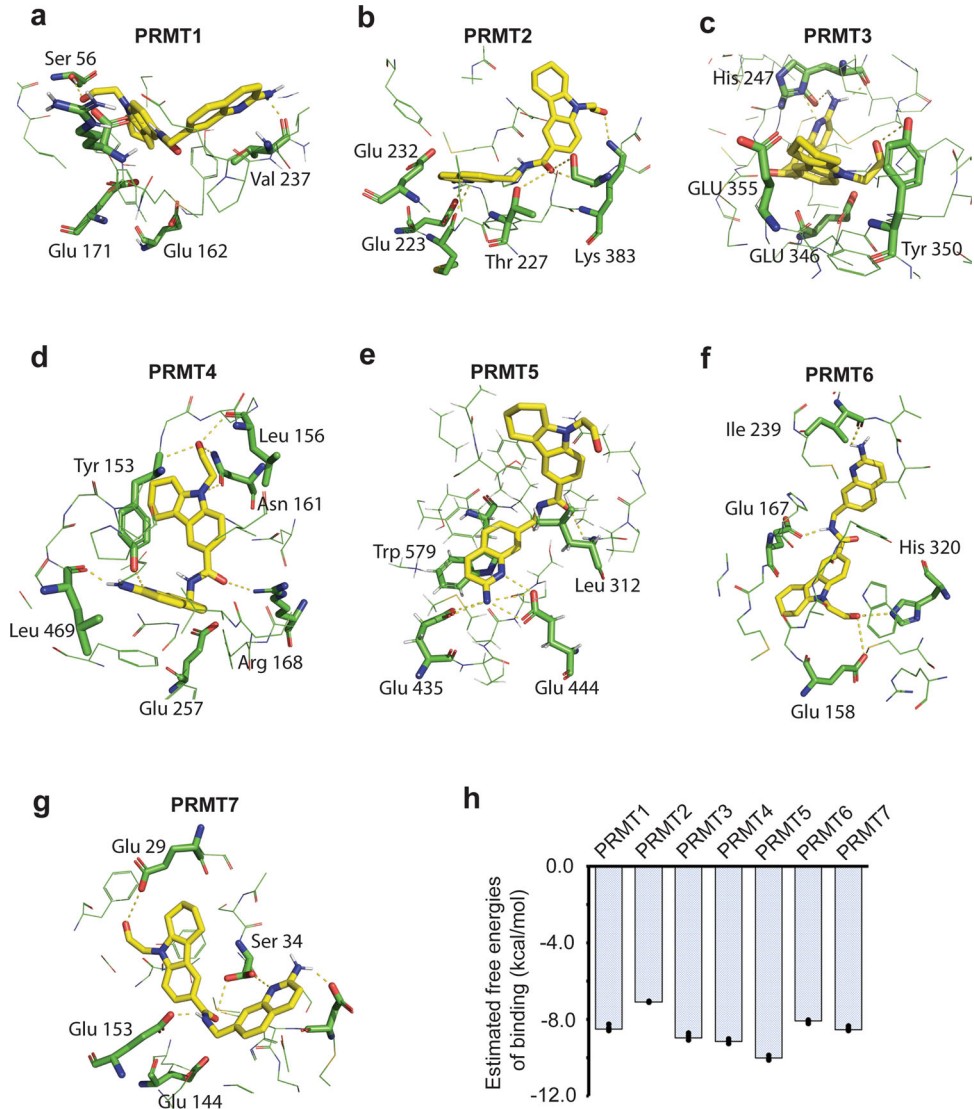

**Fig. 6 Computational analysis of 11-2F in structures of different PRMT subtypes suggests subfamily specificity.** Top pose of 11-2F in PRMT1 (PDB: 6NT7; **a**), PRMT2 (PDB: 5FUL; **b**), PRMT3 (PDB: 4QQN; **c**), PRMT4 (PDB: 3B3J; **d**), PRMT5 (**e**), PRMT6 (PDB: 3C05; **f**), PRMT7 (PDB: 4C4A; **g**). Some of the key residues contributing to interactions with the inhibitor are labeled for each. **h** Relatively binding energy levels of 11-2F to PRMT1-7. Error bars: s.d. ($n = 3$).

Similarly, the other two compounds (HWIem2104 and HWIem2109) were ranked high, and their top poses presented in Fig. 7b, c show extra interactions of their tail portions with the residues of the flexible loop (residues 292–329) when the added 4-member ring interacts with the Thr323 hydroxyl group, and the carbonyl group of its middle linker interacts with Ser310. For HWIem2109, AUTODOCK Vina predicted its top pose close to that of 11-2F (Figs. 3f vs. 7c; Supplementary Fig. 8, and Supplementary Table 7). It contains an NH- group in the 6-member alkyl ring, which forms a H-bond with Ser 310 and was predicted to be ~50 fold more potent than 11-2 F (Fig. 7d). Given that the quinazoline rings in both compounds are constrained as the head part of 11-2F, we expect that the high potency for HWIem2104 and HWIdm2109 predicted by computational analysis is probably a good indicator of their physical potency, which still awaits testing after chemical synthesis.

## Discussions

**Near-atomic resolution cryo-EM structures and computational analysis for design of high-potency inhibitors**. We showed a strategy to redesign and improve potency of an inhibitor for an oncogenic target, PRMT5, by using a cryo-EM structure to select computed compound poses. Near-atomic resolution structures by single particle cryo-EM, which are often resolved to 2.5–4.5 Å, were proposed to be useful for SBDD by multiple investigators recently[1,2,9,70,71]. Our results showed that a 3.1 Å cryo-EM map of the MTA/11-2F-bound PRMT5/MEP50 complex contains densities in the two ligand-binding pockets with clear shapes that enable the selection of top poses delivered by computational analysis through molecular docking and energy minimization. In doing so, a key step is probably the separation of the densities of two ligands from that of the protein so that the refinement of the protein model does not eclipse the optimization of the ligand models. At ~3.1 Å resolutions, two-fused rings (10-atoms) of MTA are sizeable enough for accurate modeling. The arrangement of the binding pocket residues and the accurate positioning of the adenine ring in MTA or the quinoline ring of 11-2F are important for computational analysis to gain sufficient sensitivity in differentiating the docking energy on a relative scale and allow the top cluster of compound poses to approximate true (most stable) solutions with good accuracy. For 11-2F, the density

**Table 2 Comparison of 11-2F and its designed analogs based on docking analysis using the cryo-EM structural model of PRMT5.**

| Inhibitors | Estimated binding energy (kcal/mol)(Vina) | Estimated Ki (nM) | RSMD (Å) | Some key residues in non-covalent interactions (D...H distance, Å) |
|---|---|---|---|---|
| 11-2F | −9.89 (−10.1) | 55.6 | 6.91 | Glu444:OE1 ... 11-2F:N (1.87) Glu435:OE1 ... 11-2F:N (1.87) Ser578: HG ... 11-2F:O2 (1.99) Lys333:HZ3 ... 11-2 F:O2 (1.77) Thr323:OG1 ... 11-2F:H25 (1.87) Phe327:O ... 11-2F: H26 (2.09) |
| 11-9F | −10.56 (−10.6) | 18.1 | 1.92 | Glu444:OE1 ... 11-9F: N (1.87) Glu435:O ... 11-9F:NH (1.87) Ser439:OG ... 11-9F:HN (2.1) Ser439:OH ... 11-9F:O (2.1) Phe580:NH ... 11-9F:HO (2.0) |
| HWIem2104 | −12.34 (−11.7) | 0.9 | 2.45 | Glu444:OE1 ... 2104:N (1.87) Glu435:OE1 ... 2104:N (1.87) Glu312:O ... 2104:NH (1.99) Ser310:OH ... 2104:O (1.88) Glu323:OE1 ... 2104:N (1.87) |
| HWIem2109 | −12.21 (−11.4) | 1.1 | 2.34 | Glu444:OE1 ... 2109:N (1.87) Glu435:OE1 ... 2109:N (1.87) Glu312:O ... 2109:NH (1.99) Ser310:OH ... 2109:O (1.88) Glu580:OE1 ... 2109:N (1.87) |

corresponding to the cyclo-alkylated indole rings is another determinant for its accurate positioning. Different software packages for molecular docking generate similar top poses, although their relative energetic levels vary, probably because of the same stereo chemistry parameters used for proteins and ligands. Expectedly, the results will probably be even better with cryo-EM maps of 2.0–3.0 Å resolutions, whereby individual atoms of certain multi-member rings in ligands and some of ordered water molecules at the binding pockets will become recognizable.

The strategies we tested above appear to work well for the redesign of 11-2F based on accurate modeling of the protein-ligand interactions and the computational analysis to select three different ways out of virtual modifications to decrease binding energy. The energy minimization defines the energetically favored pose with the 11-2F quinoline ring being next to the tail of MTA. One of three hits from virtual modification, 11-9F, showed significantly better potency in enzyme inhibition assay and strong MTA-synergy, close to the predicted enhancement based on relative binding energy (Table 2 and Fig. 7e). Because of their preserved head parts (quinazoline), HWIem2104 and HWIem2109 are probably going to follow the prediction and show even higher potency than 11-9F, even though they still need to be synthesized for experimental tests.

**Catalytic mechanism of PRMT5 and the MTA-inhibitor cooperativity.** Understanding the catalytic mechanism of PRMT5 may also be helpful in development of potent inhibitors. So far, all well-characterized PRMT5 inhibitors in published studies require the nucleoside component of the cofactor SAM, indicating that they are either SAM analogs or SAM-dependent molecules. Our work depicted that the cofactor binding induces a conformational change of the loop region (residues 292–329) by forming a short helix α1 (residues 320–329), which stabilizes the nucleoside via Tyr324, followed by the formation of a "lid" (residues 314–319) and a substrate-binding tunnel (residues 295–313), where Phe327 sandwiches the aromatic ring of the substrate against Trp579. These provide a direct physical connection to achieve positive cooperativity between MTA in the cofactor-binding site and the inhibitor at the substrate-binding site (Fig. 2e). MTA has a shorter tail than SAM or SAH does such that its binding induces the formation of a substrate-binding pocket sufficient to accommodate a quinoline (or quinazoline) ring well. Foreseeably, different MTA analogs might be developed to enhance the potency of the compound inhibitors, especially for those MTAP−/− cancer cells[51]. Given that the MTA and 11-2F binding sites are next to each other, a chimera harboring the key groups of the two may be prepared for the same purpose.

Our work showcased the feasibility of the proposed strategy in using cryo-EM structures of ~3.0 Å resolutions and computational analysis of compound poses for SBDD. It led to the development of a different class of substrate-competitive inhibitors that preferentially bind the PRMT5:MEP50 complex with MTA and may be used to pharmacologically exploit the PRMT5-related vulnerability in MTAP−/− cancer cells. We expect that the same or similar strategy can be tested for other cryo-EM maps of near-atomic resolutions, where the energetic calculations without ordered water molecules in the binding pockets will allow selection of the top poses by extensively sampling ligand configurations to distinguish the most stable ones from the rest. With more successful applications, this strategy will likely expand our capacity in developing new molecular therapeutics.

## Methods and materials

**Expression, purification, and characterization of PRMT5:MEP50 complex.** PRMT5:MEP50 protein complex was prepared as described by Antonysamy, S., et al.[39]. Briefly, full-length human PRMT5 (residues 1–637, NP_006100) and human MEP50 (residues 2–342, NP_077007) were co-expressed in *Sf*9 cells using a Bac-to-Bac expression system (Invitrogen). The cells were harvested and lysed. After removal of cell debris, supernatants were collected for FLAG-affinity chromatography (Sigma; A2220) and fractionated by size exclusion chromatography in a 10/300 Superose 6 column (GE Life Sciences). The purified PRMT5:MEP50 complex was characterized by SDS-PAGE and sedimentation velocity analytical ultracentrifugation (SV-AUC) using a ProteomeLab™XL-I system (Beckman Coulter; Supplementary Fig. 1). Protein samples were concentrated to 12 mg/mL in a buffer containing 10 mM HEPES at pH 7.5, 150 mM NaCl, 10% (vol/vol) glycerol and 2.0 mM DTT and were stored at −80 °C until the time of use.

**Enzymatic inhibition assay.** Enzymatic inhibition activity of 11-2F (or other compounds) was determined by MTase-Glo™ methyltransferase assay (Promega Corporation, V7602). 11-2 F was serially diluted by 5-fold from 500 μM to 6.4 nM in DMSO then added into reaction buffer (30 mM Tris-HCl at pH 7.4, 500 mM NaCl, 2 mM MgCl$_2$, 2 mM TCEP, 0.1% (wt/vol) BSA and 0.01% (vol/vol) Tween-20) with final DMSO concentration at 5% (vol/vol). The enzymatic inhibition assay was performed in a solid white low-volume 384-well plate (Greiner, #7784075) with a total reaction volume of 16 μl and in the presence of 100 nM PRMT5:MEP50 enzymes, 10 μM SAM (Sigma-Aldrich, A4377), 2 μM histone H4 peptide (1–21) as the substrate (ANASPEC, #AS-62499), and 11-2F of indicated concentrations. Reactions without enzyme were conducted as negative control and reactions

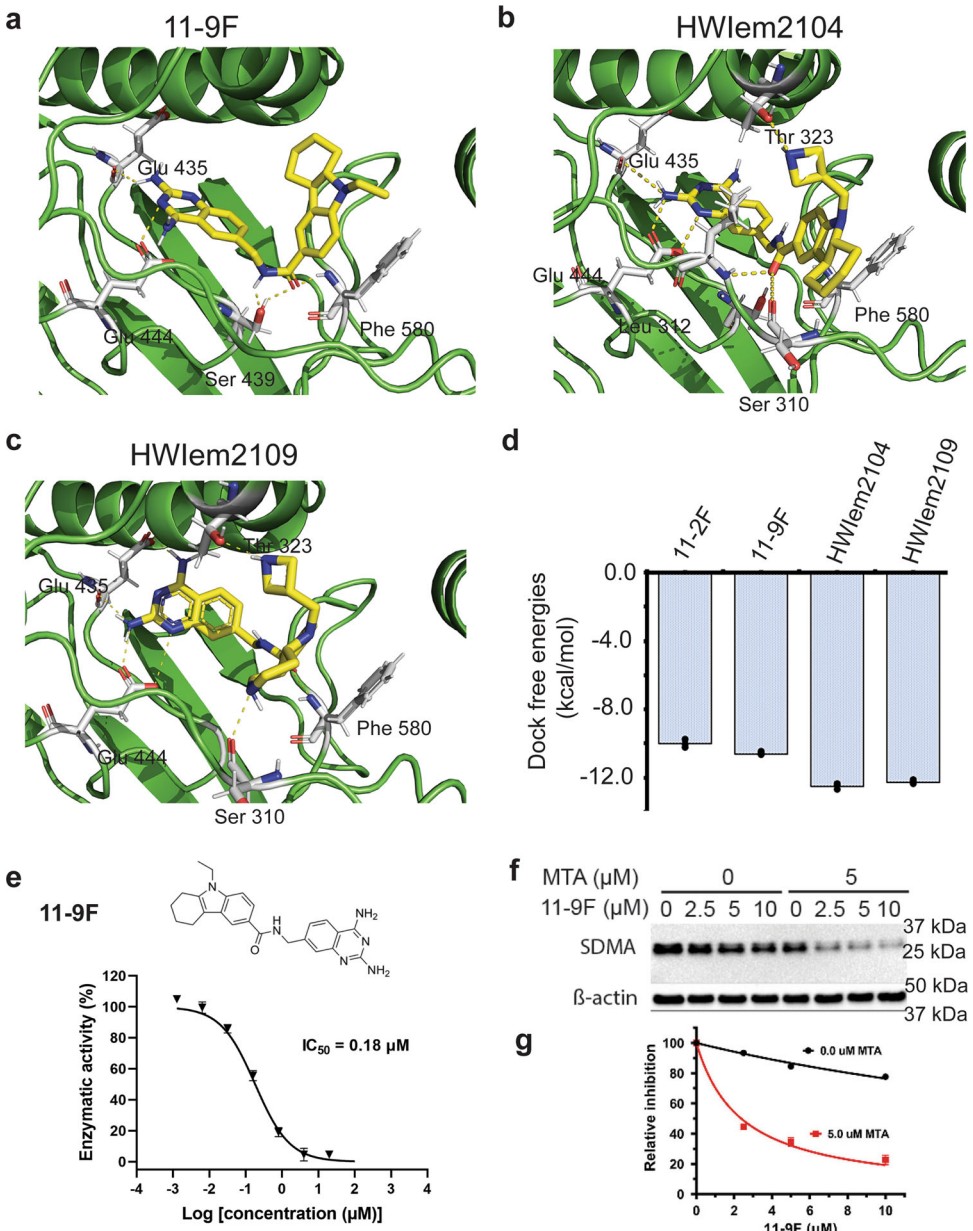

**Fig. 7 Cryo-EM SBDD yields 11-2F analogs of higher potency in PRMT5 inhibition.** Three different compounds were selected based on docking analysis. 11-9 F (**a**), HWIem2104 (**b**), and HWIem2109 (**c**) are showed in the binding pockets with key residues contributing to their stability. **d** Comparison of relative docking free energy among the four compounds. **e** The chemical structure of 11-9F (left) and its dose-dependent inhibition of PRMT5: MEP50 enzyme activity, yielding an IC50 ~ 180 nM. Error bars: s.d. (n = 3). **f** Western blotting of SDMA in cells treated with 11-9 F in different concentrations with 0 and 5.0 μM MTA. **g** Individual bands were digitized in ImageJ, calibrated against the actin bands, and then normalized against 0 μM 11-9 F in order to generate the two plots (red vs. black). Error bars: s.d., n = 3. The relative inhibition data were fitted with an equation I = 1/(1 + [L] / IC50) to IC50 of 2.4 (red trace) and 32.4 (black trace) μM, respectively. The coupling factor between MTA and 11-9F is ~10, an indicator of strong synergy.

without 11-2F were included as positive control in every experiment. Methyltransferase reaction was started by adding 4 μl of SAM/H4 substrate mixture to each well that contains 8 μl enzyme and 4 μl 11-2F pre-mixed and incubated for 10 min. The reaction was performed at room temperature for 60 min followed by the addition of 4 μl 5x MTase-Glo Reagent to produce SAH and concomitantly convert it to ADP. After shaking for 2 min and incubation at room temperature for 30 min, 20 μl room temperature MTase-Glo Detection Solution was added and mixed well before incubation for another 30 min and luminescence recording. Luminescence was measured in a Synergy Neo2 HTS multimode microplate reader (BioTek). Each data point represents the average of three replicates; the error bars represent the

standard deviations. Data are analyzed in GraphPad Prism 8. For inhibitor studies, IC50 was determined by nonlinear regression (curve fitting) using the equation for the sigmoidal dose response (variable slope).

**Surface plasmon resonance (SPR) binding study.** Binding affinity measurements were conducted in a Reichert2SPR instrument (Ametek) at 25 °C. PRMT5:MEP50 proteins were directly immobilized onto a 500,000 Da carboxymethyl dextran sensor chip at pH 5.5 using a standard amine-coupling approach. The small molecule analyte (11-2 F as an example) was injected at a flow rate of 30 μL/min of different concentrations in the running

buffer (10 mM HEPES pH 7.4, 150 mM NaCl, 3.0 mM EDTA, 0.005% v/v Tween-20, and 5% DMSO). The association and dissociation times were set to be 1 min. In the competitive binding experiment, 25 μM MTA was added into the running buffer to saturate the cofactor-binding pocket of immobilized PRMT5: MEP50 protein before the injection of the analyte. The dissociation time for 11-2 F in MTA-containing running buffer was elongated to be 3 min to achieve complete dissociation. Sensorogram data were processed using the TraceDrawer software package to calculate the equilibrium dissociation constant $K_D$, the association rate constant $k_{on}$ and the dissociation rate constant $k_{off}$.

**Cryo-EM grid preparation and imaging**. The frozen protein sample was thawed and diluted by 1:30 (vol/vol) with a buffer containing 10 mM HEPES at pH 7.5, 150 mM NaCl and 1.0% (vol/vol) DMSO to a final concentration of 0.4 mg/ml. Aliquots of 4.0 μL of the diluted proteins were applied to plasma-cleaned (Solarus, GATAN model 950 advanced plasma system), 300 mesh holey carbon-coated gold grids (Quantifoil R1.2/1.3, Electron Microscopy Sciences #Q325–AR1.3). After sample loading, the grids were incubated for 30 s in a stable environment of 10 °C and 100% humidity inside a plunge-freezing device (Vitrobot Mark III, FEI) before being sandwich-blotted for 4.0 s with a blot force of 8.0. Grids were plunge-frozen in liquid nitrogen-bathed liquid ethane and were transferred into grid-boxes and stored in liquid nitrogen before cryo-EM imaging. Grids were first screened and checked in a CM120 microscope before good grids were selected for data collection in a Titan Krios. CryoEM grids for the compound-bound PRMT5:MEP50 (PRMT5: MEP50 / 11-2 F/ MTA) were prepared in a similar way as that for the *apo* form except that the protein was incubated with the inhibitor for 30 min at a 1: 5 molar ratio of protein: inhibitor with a final protein concentration of 0.05 mg/ml right before grid preparation.

**Cryo-EM data collection for the *apo* form (PRMT5:MEP50)**. A total of 1535 micrograph movies were recorded inside a Titan Krios operated at 300 kV utilizing a Gatan K2 Direct Electron Detector (Gatan Inc.) in the super-resolution electron-counting mode at a nominal magnification of 105,000 ×. The calibrated pixel size under such conditions was 0.66 Å, and the defocus range was set in −0.75 to −3.0 μm. Each movie was collected in 12 s with an exposure time of 0.3 s per frame. The total electron dose was approximately 40 e−/Å$^2$ per movie.

**Cryo-EM data collection for compound bound form (PRMT5:MEP50:11-2F)**. A total of 4153 micrograph movies were recorded inside a Titan Krios operated at 300 kV utilizing a Gatan K3 Direct Electron Detector (Gatan Inc.) in the super-resolution electron-counting mode. The calibrated pixel size under such conditions was 0.56 Å, and the defocus range was set in −0.75 to −3.0 μm. Each movie was collected in 12 s with an exposure time of 0.3 s per frame. The total electron dose was ~40 e−/Å$^2$ per movie.

**Cryo-EM data processing for the *apo* form (PRMT5:MEP50)**. The general workflow and some intermediate results are showed in Supplementary Fig. 4. The movie stacks were motion-corrected, dose-weighted and binned by 2 using MotionCor2[72], resulting in a pixel size of 1.32 Å. Parameters for the contrast transfer function (CTF) of each movie were estimated by CTFFind4[73]. Relion3.0[74] was utilized for particle picking, extraction, 2D/3D classification, and refinement. LoG-based particle picking mode was utilized to select an initial set of

particles, which were classified and averaged as templates for subsequent template-based auto-picking. The auto-picked 597,002 particles were subjected to three rounds of 2D classification to remove junky or damaged particles, yielding a sub-set of 151,554 particles for 3D classification. An initial de novo 3D model was generated in Relion3.0. Two rounds of 3D classification were performed to identify distinct conformational states or sub-stoichiometric assemblies of the PRMT5:MEP50 complexes. Unbinned particles from good 3D classes exhibiting 1:1 stoichiometry and high-resolution structural features were re-extracted. A cleaned set of 113,466 particles were used to calculate a 3.9 Å reconstruction based on the gold-standard FSC (Fourier shell correlation)[75] with the 0.143 criterion with D2 symmetry imposed. The particle stack was then subjected to 2 iterations of CTF refinement and beam tilt correction, resulting in a 3.8 Å resolution structure. The resolution was improved to 3.6 Å by applying a soft mask around the entire protein complex during 3D auto-refinement. Bayesian polishing and another iteration of CTF refinement further improved the resolution to 3.4 Å. The refined particles were extracted and further refined in the cisTEM[76] which slightly improved the resolution to 3.17 Å. The final map was sharpened within a soft mask with an automatically calculated B-factor of −79.7 Å$^2$. The local resolution variation of the *apo* form was assessed using ResMap[77,78] and colored in Chimera[79].

**Cryo-EM data processing for the PRMT5:MEP50:11-2F/MTA complex**. The general workflow and some intermediate results are showed in Supplementary Fig. 2. The movie stacks were motion-corrected, dose-weighted and binned by 2 using MotionCor2[72], resulting in a pixel size of 1.11 Å. CTF parameters for each movie were estimated by CTFFind4[73] and 1918 micrographs with good CTF fitting were selected. CisTEM[74] was utilized for particle picking, extraction, 2D/3D classification, and refinement. The auto-picked 866,240 particles were subjected to two rounds of 2D classification to remove junky or bad particles, yielding a sub-set of 461,119 particles. Three rounds of 3D classification were performed to identify distinct conformational states or remove sub-stoichiometric assemblies of the PRMT5: MEP50 complexes using the *apo* map as a starting reference. A cleaned dataset of 213,221 particles were auto-refined in cisTEM, yielding a structure of a nominal 3.14 Å resolution based on the gold-standard FSC[75] using a threshold of 0.143 with D2 symmetry imposed. The final map was sharpened within a soft mask with an automatically calculated B-factor of −90.0 Å$^2$. The local resolution variation of the final map was assessed using ResMap[77,78].

**3D model building and refinement**. The crystal structure of an inhibitor-bound (LLY-283) form of PRMT5:MEP50 dimer (PDB: 6CKC) was used as the initial model and docked into the density map using Chimera. The model was subjected to real-space refinement using PHENIX[80] with secondary structure and geometry restraints and manually adjusted in COOT[81]. A molecular dynamics (MD)-based optimization was performed using ISOLDE[82] with the technical assistance by Dr. Tristan Croll (CCPEM)[83]. Overfitting and overinterpretation of the model were monitored by refining the model against one of the two independent half-maps and testing the refined model against the other map. The final structure was assessed in MolProbity and optimized to minimize clashes[84]. Parameters for cryo-EM data collection and modeling statistics are summarized in Table 1 for the *apo* and inhibitor-bound PRMT5 complexes.

**Molecular docking analysis with energy minimization**. Molecular docking was performed mainly by using AutoDock 4.2.6[69].

The X-ray crystal structure of human PRMT5 in complex with a substrate and a co-factor (PDB codes: 5FA5, 4X61 & 4X63), structures of different PRMTs (PDB codes: PRMT1: 6NT7, PRMT2: 5FUL, PRMT3: 4QQN, PRMT4: 3B3J, PRMT6: 3C05, PRMT7: 4C4A) and the 3D structures of MTA, SAM and SAH were retrieved from the RCSB Protein Data Bank (www.rcsb.org)[85]. All input files were prepared using the AutoDockTools (ADT) 1.5.4 package. To carry out the docking simulations, a 50 Å × 50 Å × 50 Å grid box with a lattice spacing of 0.375 Å was defined with its center at the 11-2 F position that was modeled and refined in cryo-EM map. The grid box enclosed fully the catalytic center of PRMT5. The AutoGrid program was used to construct the grid maps for energy minimization and scoring. The three-dimensional locations and orientations of the various inhibitor configurations were analyzed from 25 million configurations with randomly sampled seed parameters using a Lamarckian genetic algorithm (LGA)[86,87]. After energy minimization, the best poses from 2000 individual runs were generated, which represent 2000 typical configurations of the inhibitor and were grouped into clusters by a threshold in root mean square deviations (RMSD) of 2.0 Å. These clusters were ranked by relative binding energy. The poses of the lowest energy (optimal configurations) or the cluster of the largest sizes (suboptimal conformation) were chosen for further analyses in the PyMOL molecular graphics system (https://pymol.org/2/). Usually at least 3–5 poses were examined. More details on the three separate steps are provided in the following.

*Docking analysis using AUTODOCK.* The initial model of a ligand was generated in Chimera and went through energy minimization. The model is then docked to overlap with the ligand density in the cryo-EM map, which served as a starting point. The refined protein model based on the cryo-EM map (without the ligands) was used as a rigid model for docking of different inhibitors in AutoDock 4[12,66,69,87]. No water molecules were included. Top poses from 2000 runs were clustered into discrete groups using a Root Mean Square Distance (RMSD) threshold. Usually for each ligand we obtained 8–10 clusters and used the pose with the lowest energy from each cluster as its representative one. The top pose has the lowest docking energy ($\Delta\Delta G$) among a large number (billions or more) of different configurations.

*Comparison of results from three different software packages.* Similarly, the same scheme was implemented in AutoDock Vina and an online server, SwissDock[65,68,88,89]. The top poses for the same ligand from three different software packages against the same protein structural model were very similar, if not the same, suggesting that the large number of starting positions sampled by this protocol were able to reflect almost all possible configurations of the ligand within a small range of errors.

*Refinement of the top pose against the cryo-EM density of the ligand.* The refinement of the top pose of a ligand was performed by simplified all-atomistic molecular dynamics (MD) calculations implemented in ISOLDE[67]. The refinement may slightly improve the model for better match with the density. For MTA, the refinement introduced small changes of the ligand model. For 11-2 F, the refinement led to a 180° flipping of the quinoline ring around a C-C bond connected to its 6th position.

**Testing MTA-synergy of 11-9F in cultured cells.** MDA-MB-231 cells were seeded at $1.0 \times 10^6$ cells/dish and allowed to adhere overnight. The cells were then treated with indicated doses of the test compounds (11-9F) in complete growth medium for 96 h. The cells were harvested and lysed in cold RIPA lysis buffer (Pierce #89900) containing protease inhibitor cocktail and

phosphatase inhibitor cocktail (Pierce #78441). The protein concentrations were determined using the BCA Protein Assay kit (Pierce #23227). After being boiled in loading buffer at 95 °C for 10 min, equivalent amounts of proteins were resolved in a 10% SDS-PAGE gel, and then were transferred to a PVDF membrane (Bio-Rad #1620174) at 110 V for 1 h. The membrane was treated in a blocking buffer (Pierce #37530) and incubated with corresponding primary antibodies: anti-SDMA (1:1000, Cell Signaling #13222 S), rabbit anti-ADME-R (1:1000, Cell Signaling #13522 S), and anti-β-actin (1:1000, Cell Signaling #4970 S) in the blocking buffer at 4 °C overnight. After three washes with TBST buffer (Pierce #28352), the blots were incubated with goat anti-rabbit secondary antibody (HRP-conjugated, 1:10000, Invitrogen #31460) in the blocking buffer for 1 h at RT. After three washes with the TBST buffer, the membrane was analyzed using ECL Western blotting detection reagents (Pierce #34096) and imaged in a ChemiDoc MP imaging system (Bio-Rad). Digitization of the blot bands was done in ImageJ (NIH; Supplementary Fig. 9). Individual bands were normalized against the actin loading control before being compared against the ones with no inhibitor.

**Compounds 11-2F and 11-9F.** The two compounds were synthesized, purified and verified in Dr. Li's lab. After HPLC purification the compound purity was more than 95%. The molecular mass and structural details were verified by mass spectrometry and NMR analysis[63]. Their IUPAC names are 11-2F: N-[(2′-aminoquinoline)-7′-methyl]-[9-(2-hydroxyethyl)-5,6,7,8-tetrahydro-carbazole]-3-carboxamide, and 11-9 F: N-[(2′,4′-diaminoquinazoline)-7′-methyl]-[9-ethyl-5,6,7,8-tetrahydro-carbazole]-3-carboxamide.

**Statistics and reproducibility.** We used two-tailed Student's *t* test to compare experimental data from separate experiments done in parallel or in repeated fashions. Data from three or more repeated experiments were included to enhance reproducibility. For structural modeling and comparison, the statistical analysis was performed in different software packages.

**Reporting summary.** Further information on research design is available in the Nature Research Reporting Summary linked to this article.

## Data availability

Cryo-EM density maps of the *apo* and 11-2F-bound forms of human PRMT5:MEP50 have been deposited in the EM data bank under the accession codes of EMD-20764 and EMD-27078, respectively. Atomic coordinates for the molecular models have been deposited into the protein data bank under the accession codes 6UGH and 8CYI, respectively.

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

## Acknowledgements

This research was, in part, funded by National Institute of Health Grants (R01GM093271, R21GM131231, and R01GM111367 to Q.-X.J.), a Cystic Fibrosis Foundation grant JIANG15G0 (to Q.-X.J.), a NIH/NCI grant R01CA212403 (to C.L.) and the UFHCC Shands fund (to C.L.), the SEM4 consortium at Florida State University (U24GM116788) with Dr. Kenneth Taylor as the PI and Q.-X.J. as one of the co-PIs, the NIH-funded Midwest consortium for High-resolution Cryo-EM at the Purdue University (U24GM116789) with Dr. Wen Jiang as the contact PI and Q-X.J. as one of the co-PIs, the National Cancer Institute's National Cryo-EM Facility at the Frederick National Laboratory for Cancer Research under contract HSSN261200800001E. Part of the startup package (to Q-X.J.) from the IFAS of UF was used to support this work. Besides, the cryo-EM studies in the Jiang lab have utilized outside facilities at the Case Western Reserve University (thanks to Dr. Kunpeng Li), the National Center for CryoEM Access and Training (NCCAT) and the Simons Electron Microscopy Center located at the New York Structural Biology Center (NYSBC) under the support of the NIH Common Fund Transformative High Resolution Cryo-Electron Microscopy program (U24GM129539) and by grants from the Simons Foundation (SF349247) and NY State Assembly Majority. PRMT5 and MEP50 plasmids were shared by Dr. Xiaojie Zhang from the Haitao Li Lab at Tsinghua University. We are grateful to Drs. Kenneth Taylor, Xiaofeng Zheng, Nilakshee Battacharya, and others at FSU, Drs. Wen Jiang, Thomas Klose and others of the team at Purdue, and Drs. Ulrich Baxa, Thomas Edwards, Adam Wier and others at NCEF of NCI for their technical assistance in data collection. Q.-X. J. and G.Y. thank Dr. Tristan Croll for helping with the use of ISOLDE.

## Author contributions

C.L. and Q.-X.J. initiated the project. Q.-X.J designed major steps of the studies and discussed with the rest of the group to implement them. W.Z. and G.P.Y. contributed to cryo-EM grid preparation, grid screening, data collection, and analysis as well as manuscript preparation. W.Z. conducted protein expression and purification, enzymatic assay and SPR binding assay. X.Y. performed compound design and synthesis and purification. F.Q helped the biophysical analysis with Q.-X.J. C.L. helped Q.-X.J. to supervise the project and participated in revising the manuscript. Q.-X.J. supervised all parts for cryo-EM, including grid preparation and evaluation, data collection, data analysis, model building, computational analysis, and molecular interpretation. Q.-X.J. wrote the manuscript based on earlier writings by G.P.Y and W.Z. and conducted revisions together with the rest of the group.

## Competing interests

The authors declare no competing interests.
