## [Peer Review File · Communications Biology]

Reviewers' comments:

Reviewer #1 (Remarks to the Author):

In this manuscript, Yadav et al. described the first-in-class proof of concept, Cryo-EM structure-guided selection of computed ligand pose to enhance ligand potency. Generally, resolution of cryo-EM structures ranging from 2.5 to 4.5 Å make them limited utility in structure-based drug design. However, combination of the molecular docking with cryo-EM tec. could open a new phase. In the manuscript, Yadav et al discovered the novel MTA-synergistic PRMT5 inhibitor hit compound 11-2F by virtually screening, then solved the PRMT5-MTA-11-2F cryo-EM complex to select the reasonable docking ligand pose and explained the mechanism of interaction between PRMT5 and hit compound in the presence of MTA, further led to structure-based design of three potentially more potent PRMT5 inhibitors. However, several comments below should be addressed before publication.

1. In manuscript, Page 8, just comparing the affinities of 11-2F between to PRMT5-MEP50 and to PRMT-MEP50-MTA is not improper, at least not enough. The affinity to PRMT5-MEP50 in the presence of SAM should be also compared. That data could also validate the conclusion in Paragraph 2, Page 14.

2. MRTX9768, as a reported PRMT5/MTA synergistic inhibitor, its chemical structure and its co-crystal structure with PRMT5/MTA have been disclosed. https://www.mirati.com/wp-content/uploads/AACR-2021_Fragment-based-discovery-of-MRTX9768-a-synthetic-lethal-based-inhibitor-designed-to-bind-the-PRMT5-MTA-complex-and-selectively-target-MTAPDEL-tumors_Final.pdf. It is interesting to see the differences between MRTX9768 binding pose to PRMT5/MTA and 11-2F binding pose to PRMT5/MTA described in this manuscript.

3. In Page 6, first line, another new class of PRMT5 inhibitor should be included. doi.org/10.1021/acs.jmedchem.1c00507

4. In Page 14, paragraph 2, EPZ015666 is not a JNJ inhibitor.

5. In Page 10, paragraph 2, Table S3 should be Table S2

6. In Page 11, paragraph 2, Table S4 and S5, should be Table S3 and Table S4, respectively.

Reviewer #2 (Remarks to the Author):

In the manuscript by Gaya P. Yadav and et al., authors describe cryoEM structure-guided selection of inhibitors to enhance potency in MTA-bound PRMT5. In this study, a lead compound, 11-2F was selected based virtual screen and the K_i is 13.6 μM but can be enhanced to ~ 82 nM in presence of 25 μM MTA. To understand this synergistic inhibition, authors determined a 3.1Å cryoEM of 11-2F/MTA bound PRMT5:MEP50 applied a D2 symmetry during refinement. Considering the "low" resolution, authors computed different ligand poses by AUTODOCK and found a better docking for MTA and 11-2F by MD calculation. One of the highlights authors claim is that this computer-aid ligand docking is suitable to "low resolution" cryoEM structure. In addition, an apo PRMT5:MEP50 with 3.2 Å was determined. Structural comparison with apo and MTA/11-2F bound form, authors found MTA and substrate (11-2F?) induced some local conformation changes. Later, author also rationalize the synergy between MTA and 11-2F by comparing with MTA/H4-bound X-ray structure and found the cofactor binding pocket is pushed upwards by 2Å. Author further predict subtype specificity of 11-2F among PRMTs and claim 11-2F is PRMT5-specific. Lastly, author analyze their structure and rationalize the key interaction for inhibition. Based on that, analogs of 11-2F were predicted to have more potent inhibition. One of them, 11-9F was synthesized and tested to had 4-5 folds more potency compared to 11-2F.

PRMT5 is an emerging drug target for various cancers. This work is very interested and provides some insights in PRMT5 drug discovery.

However, parts of this manuscript is confusing and require further clarification.

1. First of all, authors emphasized that the low resolution (2.5-4Å) of cryoEM structure is challenging to structure-based drug discovery compared to high resolution of X-ray structure. They proposed a computer-aid ligand modeling strategy including molecular docking, energy

minimization and pose selection and use this PRMT5 as proof-of-concept. Although the cryoEM indeed provide lower resolution but the map quality is, in general better than those same resolution from protein crystallography. The actual issue is the local resolution of cryoEM map varies a lot due to the nature of this method whereas resolution crystallography map is generally similar. Often for the resolution of region of interest is much lower than the averaged resolution and is too poor to model the ligand. In this case, the computer-aid strategy can be really helpful. HOWEVER, in this case, the active site is actually much better than reported and near to 2.4Å (estimated from figure 2B). To be fair, authors should report the local resolution of binding pocket region.

For a 2.4Å cryoEM map whose map quality is better than 2.4Å X-ray map, I am not sure if it is necessary to use this computer-aid strategy. From the density map of ligand region shown in Figure 3B, F and H, it seems to me that the conventional manual model building approach using coot/phenix would get the similar result.

2. Regarding to cryoEM structure-guided drug discovery, the issue is that low resolution can not reveal 1) the accurate position of side chain and 2) critical water molecules for ligand-enzyme interaction. It is not clear to me if the molecular docking and energy-minimization consider the local conformation changes of active-site side chains and water molecules upon the ligand binding. As the author showed the active site region change upon MTA binding.

3. In the introduction, authors mentioned PRMT5 inhibitor is required for MTAP-deficiency cancer cells whose MTA is accumulated. It is puzzling to me because MTA indeed inhibit PRMT5 so why a potent PRMT5 (or different class of PRMT5) is required.

4. In the structural basis for synergy between MTA and 11-2F, authors compared the MTA/H4 bound crystal structure and MTA/11-2F cryoEM structure and found a 2Å shift of co-factor binding domain. Based on that, they propose this movement enhances MTA binding and synergistically enhance 11-2F binding. However, authors did not consider this 2Å may be due to the protein crystal packing force. Furthermore, why MTA binding enhancement can affect 11-2F? In the binding experiment, authors only provide the binding improvement of 11-2F from 13.6µM to 82nM (>160-fold change) but not binding data from MTA. It seems no direct interaction between MTA and 11-2F. It is also unusual that authors only reveal MTA and 11F-2 individually but not how these two compounds are "occupied together" (only vaguely shown in Figure 5) while they spent a lot of effort in describing the docking part.

Minor Concerns.

1. Page 7. Why do authors reason that near 3Å resolution may have sufficient structural constraints?

2. Page 8, It seems that 11-2F is a newly discovered PRMT5 inhibitor. How virtual screening identify this compound is not very clear. For example, the resulting pharmacophore was modified manually based on crystal structure

3. In addition, is 11-2F occupied the substrate binding pocket? This should be clarified somewhere since page 13 line two brings up no clear densityfor MTA and substrate.

4. Page 9, in the method, the D2 symmetry is applied and this should be mentioned in the result.

5. Page 9, the complex is prepared by mixing 500 µM MTA, 100 µM of 11-2F and 0.09 µM (0.4 mg/ml)PRMT5:MEP50 but in the supplementary, it state 1:5 protein: inhibitor molar ration and final protein concentration to 0.05mg/ml.

6. Page 12, a type of MTA.. 11-2F/MAT

7. Supplementary Figure S2 and S4 should contain more detailed info. For example, when D2 symmetry is applied and the improved resolution. The FSC for final refinement is barely readable.

Reviewer 1

1. In manuscript, Page 8, just comparing the affinities of 11-2F between to PRMT5-MEP50 and to PRMT-MEP50-MTA is not improper, at least not enough. The affinity to PRMT5-MEP50 in the presence of SAM should be also compared. That data could also validate the conclusion in Paragraph 2, Page 14.

Done as suggested.

2. MRTX9768, as a reported PRMT5/MTA synergistic inhibitor, its chemical structure and its co-crystal structure with PRMT5/MTA have been disclosed. <https://www.mirati.com/wp-content/uploads/AACR-2021-Fragment-based-discovery-of-MRTX9768-a-synthetic-lethal-based-inhibitor-designed-to-bind-the-PRMT5-MTA-complex-and-selectively-target-MTAPDEL-tumors-Final.pdf>. It is interesting to see the differences between MRTX9768 binding pose to PRMT5/MTA and 11-2F binding pose to PRMT5/MTA described in this manuscript.

We are aware of this information and would like to compare our data with the reported inhibitor, but unfortunately, the structural information for the above-mentioned inhibitor (MRTX9768) and a related one (MRTA1719) in a recent publication is not available in wwPDB, which makes it impossible to do structural comparison. We updated the text (Lines 120-124).

3. In Page 6, first line, another new class of PRMT5 inhibitor should be included. doi.org/10.1021/acs.jmedchem.1c00507

We added the fourth class on the substrate adaptor proteins (SAPs) and cited two relevant papers (Lines 115-117).

4. In Page 14, paragraph 2, EPZ015666 is not a JNJ inhibitor.

Corrected. Thanks.

5. In Page 10, paragraph 2, Table S3 should be Table S2

Corrected.

6. In Page 11, paragraph 2, Table S4 and S5, should be Table S3 and Table S4, respectively.

Corrected.

Reviewer 2

1. First of all, authors emphasized that the low resolution (2.5-4Å) of cryo-EM structure is challenging to structure-based drug discovery compared to high resolution of X-ray structure. They proposed a computer-

aid ligand modeling strategy including molecular docking, energy minimization and pose selection and use this PRMT5 as proof-of-concept. Although the cryo-EM indeed provide lower resolution but the map quality is, in general better than those same resolution from protein crystallography. The actual issue is the local resolution of cryo-EM map varies a lot due to the nature of this method whereas resolution crystallography map is generally similar. Often for the resolution of region of interest is much lower than the averaged resolution and is too poor to model the ligand. In this case, the computer-aid strategy can be really helpful. HOWEVER, in this case, the active site is actually much better than reported and near to 2.4Å (estimated from figure 2B). To be fair, authors should report the local resolution of binding pocket region.

Please see the second point in the top part.

2. Regarding to cryo-EM structure-guided drug discovery, the issue is that low resolution cannot reveal 1) the accurate position of side chain and 2) critical water molecules for ligand-enzyme interaction. It is not clear to me if the molecular docking and energy-minimization consider the local conformation changes of active-site side chains and water molecules upon the ligand binding. As the author showed the active site region change upon MTA binding.

We agree with the limitations. Due to the resolutions, water molecules were not resolved clearly in the binding pocket. All water molecules were removed from the models used in the docking calculations. We treated protein molecule in a fixed state during the calculations while the ligand was free to rotate at all rotatable bonds to sample different configurations of the inhibitor, which is routinely done in molecular docking calculations. Theoretically, the partial charges of the ligand are only partially shielded by the polarized water molecules that might sit between the ligand and the protein so that the general energy terms in the presence of structured water molecules would be very close to those without them. We added the rigidity part in the method description (lines 548-551).

Our results are encouraging because the docking calculations were able to tell apart the head and tail parts of the ligand in the binding pocket, and the search of redesigned ligands led to hits of better potency. We suggested that this strategy needs more tests in other systems before being more generally applicable.

3. In the introduction, authors mentioned PRMT5 inhibitor is required for MTAP-deficiency cancer cells whose MTA is accumulated. It is puzzling to me because MTA indeed inhibit PRMT5 so why a potent PRMT5 (or different class of PRMT5) is required.

We revised the text to make this point clearer (Lines 86-90). PRMT5 is upregulated in many human cancer types, including lymphomas, lung cancer, breast cancer and colorectal cancer. The MTA accumulation sensitizes the cancer cells for PRMT5 inhibition such that the MTA-synergic inhibition of PRMT5 works much better in MTAP^{-/-} cancer cells than MTAP^{+/+} ones.

4. In the structural basis for synergy between MTA and 11-2F, authors compared the MTA/H4 bound crystal structure and MTA/11-2F cryoEM structure and found a 2Å shift of co-factor binding domain. Based on that, they propose this movement enhances MTA binding and synergistically enhance 11-2F binding. However, authors did not consider this 2Å may be due to the protein crystal packing force. Furthermore, why MTA binding enhancement can affect 11-2F?

In the binding experiment, authors only provide the binding improvement of 11-2F from 13.6 μ M to 82nM (>160-fold change) but not binding data from MTA. It seems no direct interaction between MTA and 11-2F. It is also unusual that authors only reveal MTA and 11F-2 individually but not how these two compounds are “occupied together” (only vaguely shown in Figure 5) while they spent a lot of effort in describing the docking part.

Yes, thanks for raising this point. We did consider whether the 2 Å shift might have come from crystallization packing force. When we compared the X-ray structure and our ligand-bound PRMT5 structure by global alignment, we found that the RMSD is less than 1.0 Å (Fig. 2C), meaning that the global effect from crystal packing is negligibly small. The changes in the binding pocket are local.

Fig. 2E was revised to show the MTA and 11-2F together.

Minor Concerns:

1. Page 7. Why do authors reason that near 3Å resolution may have sufficient structural constraints? It is because of the better quality of map with the stably bound inhibitor, even though the binding pockets are usually at the periphery.

Our reasoning was based on the map quality and phase accuracy of cryo-EM maps being slightly better than the X-ray maps at similar resolutions. Still, structures of ~3Å show basic contours of the main and bulky side chains well, and the atomic models are inferred during model refinement. Even though the map quality of the ligand (11-2) is not better than what the average resolution suggests because the binding pocket is at the surface, we reasoned that the confidence of the atomic positions in the structures of such resolutions may be good enough for estimating the interaction energy between different ligand poses and the protein model when partial charges are used for coarse-grain calculations. Our test suggests that our strategy may be worthwhile, although more tests are still needed.

2. Page 8, It seems that 11-2F is a sulfonyl pharmacophore was modified manually based on crystal structure

The detailed procedure for the virtual screening was reported by Alinari et.al in 2015, which was cited in the supplementary info. We added it to the main text in revision (Line 164).

3. In addition, is 11-2F occupied the substrate binding pocket? This should be clarified somewhere since page 13 line two brings up no clear density for MTA and substrate.

Yes, 11-2F is in the substrate-binding pocket. In revision, we have specified such details when describing the *apo* structure without densities for MTA and 11-2F (lines 279-289).

4. Page 9, in the method, the D2 symmetry is applied and this should be mentioned in the result.

Done as suggested. Symmetry information and resolutions in different stages of data analysis in the supplementary figure S2 were added.

5. Page 9, the complex is prepared by mixing 500 μM MTA, 100 μM of 11-2F and 0.09 μM (0.4 mg/ml) PRMT5:MEP50 but in the supplementary, it states 1:5 protein: inhibitor molar ration and final protein concentration to 0.05mg/ml.

Revised to be clear because the stock solutions were used to mix the proteins with inhibitors.

6. Page 12, a type of MTA. 11-2F/MAT

Corrected. Thanks.

7. Supplementary Figure S2 and S4 should contain more detailed info. For example, when D2 symmetry is applied and the improved resolution. The FSC for final refinement is barely readable.

Done as suggested. More detailed information was added to the Supplementary Figures S2 and S4. Both figures were regenerated to accommodate reviewer's concerns and make the FSC graphs better in readability.

Please let us know if more information is needed.

Best regards,

Qiu-Xing

Chenglong

Reviewers' comments:

Reviewer #1 (Remarks to the Author):

Basically, I am satisfied to see the reply from authors. Publish as it is.

Reviewer #2 (Remarks to the Author):

Despite the importance of PRMT5 inhibitor discovery,

1. one of the highlights in the manuscript (or previous version) is the computer-aid ligand docking is suitable to "low resolution" cryoEM structure. The local resolution of the core is 2.6-2.8 angstrom which is way better than most cryoEM map. In addition, it has been known that the cryoEM map quality is better than the one from X-ray crystallography at the same resolution due to the phase issue. In the response letter, author claimed that they have difficulties in determined two-ring or three-ring in the density so they rely on docking. It is normal that the compound or even amino acid side chain position cannot simply be revealed at resolution of 2.6-3.2angstrom, the structural biologist can determine the final pose with the help of geometry restrain and surrounding interaction. Author did not clearly demonstrate the issue of 11-2F fitting into the map in this revision and the importance. Furthermore, in the MTA show in figure 3B, the only one pose fits into the density so I am wondering why the other two are required. Frankly speaking, some of 11-2F posed in the Figure s3 would not fit into cryoEM map at all.

More importantly, author admitted "this strategy needs more experimental tests in other cases before it can become generally applicable in the future". In the abstract/introduction/discussion, author still strongly imply their strategy can be the solution of drug discovery using "near atomic resolution" (3.5-4.5 angstrom) cryoEM structure as they claim "it became interesting to test the combination of virtual screening, molecular docking and energy minimization of ligands with "near-atomic resolution cryo-EM structures to enhance ligand potency. We started this direction in 2015 and selected human protein arginine methyltransferase 5 (PRMT5) as a target." This really confuse me as their map quality is 2.6 angstrom at the core and author admit that more tests are needed in the rebuttal letter.

2. The other highlight of the work is that by docking, authors are able to do virtual modification. One of them, 11-9F (IC50 ~180 nM) showed 4 time better than 11-2F (IC50 730nM. First of all. 4 time is not significantly better potency and not sure why author claim 5-time potency in page 18. More importantly, this inhibition assay is in the absence of MTA. Authors mentioned the binding of 11-2F is 82nM in the presence of MTA compared to 13.6uM in absence of MTA and the structure and docking is based on the presence of MTA structure, the comparison should be done in the presence of MTA.

3. Lastly, the author claim MTA competes with SAM and reduces the efficacy of many SAM-dependent class 2 inhibitors. They claimed that "class 2 inhibitors working in synergy with MTA" which is highlight of 11-2F because the binding affinity improves in the presence of MTA. MTA itself is already an inhibitor of PRMT5 so PRMT5 inhibitor sensitive the cancer cell in MTAP-/- cell. But author failed to address my previous question "It is puzzling to me because MTA indeed inhibit PRMT5 so why a potent PRMT5 (or different class of PRMT5) is required."

Dear Editor,

We would like to thank reviewer #2 for the constructive critiques that helped us improve the manuscript. In the next we addressed the five points one-by-one.

1). one of the highlights in the manuscript (or previous version) is the computer-aid ligand docking is suitable to “low resolution” cryoEM structure. The local resolution of the core is 2.6-2.8 angstrom which is way better than most cryoEM map. In addition, it has been known that the cryoEM map quality is better than the one from X-ray crystallography at the same resolution due to the phase issue. In the response letter, author claimed that they have difficulties in determined two-ring or three-ring in the density so they rely on docking. It is normal that the compound or even amino acid side chain position cannot simply be revealed at resolution of 2.6-3.2angstrom, the structural biologist can determine the final pose with the help of geometry restrain and surrounding interaction. Author did not clearly demonstrate the issue of 11-2F fitting into the map in this revision and the importance. Furthermore, in the MTA show in figure 3B, the only one pose fits into the density so I am wondering why the other two are required. Frankly speaking, some of 11-2F posed in the Figure s3 would not fit into cryoEM map at all.

More importantly, author admitted “this strategy needs more experimental tests in other cases before it can become generally applicable in the future”. In the abstract/introduction/discussion, author still strongly imply their strategy can be the solution of drug discovery using “near atomic resolution” (3.5-4.5 angstrom) cryoEM structure as they claim “it became interesting to test the combination of virtual screening, molecular docking and energy minimization of ligands with “ear-atomic resolution cryo-EM structures to enhance ligand potency. We started this direction in 2015 and selected human protein arginine methyltransferase 5 (PRMT5) as a target.” This really confuse me as their map quality is 2.6 angstrom at the core and author admit that more tests are needed in the rebuttal letter.

As the reviewer knows well, resolution determination in single particle cryo-EM is not as exact as that for crystallographic data. As we mentioned in the last revision, the ResMap local resolutions are relative and are used to show the expected resolution variation --- the resolution in the core is often better than that at the periphery. Based on experiences, ResMap tends to overestimate the resolutions than that from the gold-standard FSC by 0.1-0.4 Å. We thus elected to be conservative in using the FSC estimate in order to avoid overstating the achieved resolutions to be 2.6-2.8 Å.

To help the visualization and explanation, we added a panel in the supplementary Figure S3E to show that rotating the two parts of 11-2F by 180 degrees still allows the ligand to be relaxed and match the density well, which demonstrates the power of energy minimization in finding the best

pose(s) with the head part of 11-2F (quinoline ring) inserted into the inner portion of the binding pocket.

The Fig. 3B was purposefully prepared to demonstrate the quality of the docking analysis of MTA and the need of cryo-EM map-based selection by showing its top three poses whose predicted binding energy levels are close to each other. Yes, one of the poses is clearly better.

As pointed out by the reviewer, it is true that some of the 11-2F poses in the supplementary Figure S3D from docking analysis do not match with the cryo-EM density well, probably because the binding pocket and 11-2F are not tightly packed against each other, leaving space for cryo-EM map-based selection among different poses and redesigning the ligand by introducing additional chemical groups.

We revised the abstract (lines 26-39) and lines 61-74, 124-140, 430-438 of the main text to minimize possible confusions as pointed out by the reviewer.

2. The other highlight of the work is that by docking, authors are able to do virtual modification. One of them, 11-9F (IC₅₀ ~180 nM) showed 4 time better than 11-2F (IC₅₀ 730nM. First of all, 4 time is not significantly better potency and not sure why author claim 5-time potency in page 18. More importantly, this inhibition assay is in the absence of MTA. Authors mentioned the binding of 11-2F is 82nM in the presence of MTA compared to 13.6uM in absence of MTA and the structure and docking is based on the presence of MTA structure, the comparison should be done in the presence of MTA.

Thanks for catching the discrepancy of 4 or 5 times better in potency. Due to variation ranges among the experimental data points, it is probably better to be conservative and report ~4 fold (or slightly over four fold) increase in IC₅₀ for 11-9F, which is still significant when we tried to enhance the IC₅₀ from ~ 1.0 micromolar to ~150 nM range. We changed the main text and abstract to be consistent. Notably, such a change is close to the predicted fold of changes by energy analysis (Table 1).

Because the test for better potency of an inhibitor relies on the inhibition of the enzymatic activity before cell-based studies, we compared the IC₅₀ of 11-9F and 11-2F directly, instead of the SPR-based binding. The binding affinity is not a direct indicator for ligand potency in enzyme inhibition.

Our focus in this paper is not on developing the compounds and their analogs for killing cancer cells. We thus only presented data for feasibility test and showed that combining computational analysis of ligand poses with cryo-EM data-based selection appears suitable for achieving more potent PRMT5 inhibition. A more detailed analysis of 11-9F and its analogs as anti-cancer drugs will be presented in a separate manuscript (under preparation by X.Y. et al).

3. Lastly, the author claim MTA competes with SAM and reduces the efficacy of many SAM-dependent class 2 inhibitors. They claimed that “class 2 inhibitors working in synergy with MTA” which is highlight of 11-2F because the binding affinity improves in the presence of MTA. MTA itself is already an inhibitor of PRMT5 so PRMT5 inhibitor sensitive the cancer cell in MTAP^{-/-} cell. But author failed to address my previous question “It is puzzling to me because MTA indeed inhibit PRMT5 so why a potent PRMT5 (or different class of PRMT5) is required.”

For the sensitization of MTAP^{-/-} cancer cells to PRMT5 inhibition, we cited a published work by Mavrakis, et. al. (2016), entitled “Disordered methionine metabolism in MTAP/CDKN2A-deleted cancers leads to dependence on PRMT5. *Science*, 351(6278), 1208-1213.” It is #31 in the reference list after the main text.

The authors of this paper suggested that “MTA in cells creates a partially inhibited enzyme state and thus significantly increases their vulnerability to further PRMT5 inhibition”. “A model in the above figure depicts the collateral vulnerability in MTAP⁻ cells in comparison with wildtype (MTAP⁺) cells.

PRMT5 is essential for cell viability and its complete inactivation is not tolerated by most cells". Accumulation of MTA in the MTAP- cells to varied extents makes them more sensitive to PRMT5 inhibition than the MTAP+ cells. This model suggests that the inhibitory effect caused by accumulated MTA is not enough to kill the cells, but MTA-synergic agents further inhibiting PRMT5 activity to a level below cell viability threshold will make the MTAP-/- cells much more sensitive to these inhibitors. Based on this model, the true test of the compound potency as an anti-cancer drug must be performed by comparing viability differences between MTAP+/+ and MTAP-/- cancer cells, and will be done in future publications (such as the manuscript being prepared by X.Y. et al).

We revised the text in lines 78-83, 111-114 and 124-130 to make this point clearer.

4. Reviewer #2 comments:

As I mentioned those are my major concerns. There are some minor ones. For example, the 2 angstrom shift upon inhibitor binding they observed should have a better description and illustration.

To make it easier to appreciate the ~2.0 Å shift, we modified Fig. 5B by removing the stereo view, zooming into the binding pocket of MTA, and adding a vertical arrow to mark the physical shift.

We revised the text in lines 284-296 to make a more clearcut description.

5. If the focus is shifting to PRMT5 drug discovery which is very important, some more experiments may be required. For instance, the inhibitor specificity can be easily tested. The residues surrounding to the inhibitor should be mutated for the inhibition test as well.

We do not want to shift the focus to the PRMT5 drug discovery in this manuscript because the real tests for the efficacy of an anti-cancer drug need to be done in different cancer cells or animal models and in a comprehensive fashion. Further studies of 11-9F and its analogs will compare them in enzyme activity assays, cell permeability assay, cell-survival assays, etc. These will be reported in a separate manuscript.

Please let me know if there are other questions.

Best regards,

Qiu-Xing

Reviewers' comments:

Reviewer #2 (Remarks to the Author):

In the second revision of manuscript by Wei Zhou et al (the order of first author has been changed since revision), authors insist their major focus is not on developing PRMT5 inhibitor for cancer treatment, which would be the other manuscript. They present data for feasibility test and showed the combining computation analysis of ligand poses with cryoEM data-based selection appears suitable for achieving more potent PRMT5 inhibitor.

Although I think the greater impact is for PRMT5 inhibitor in cancer treatment, I respect their decision. However, authors describe the low-resolution issue of cryoEM and claim their approach can apply to "near-atomic resolution cryoEM structure work for better small molecule therapeutics". In this is true, this can be an interest to the audience of Communications Biology. However, I have raised my concerns regarding to that claim since my very first review report. Despite authors agree my concerns every time and stated the revised the abstract and main text to minimize possible confusion, I have checked the current version and previous version. Author kept this claim all the time. What I can only find the change of wording (see attachment for comparison) but no conceptual change to address my concern.

For their approach to other cryoEM work, the following are my concerns which do not address in the manuscript.

1. Concern to apply this "high resolution" example to other "low resolution" case.

The resolution of this work is 3.1 angstrom and for the active part is even better (~2.6-2.8 angstrom) because the active site is buried inside. First of all, the near-atomic resolution is >3.5 angstrom. Among 16694 cryoEM single-particle structures deposited in EMDb, only 2928 (less than 17.5%) structure has better than 3.1 angstrom resolution, 1218 (~7%) structures (has better than 2.8 angstrom resolution and 612 (3.6%) structures has better than 2.6 angstrom resolution. The PRMT5 case is a high resolution structure and, luckily, the resolution of active site is considered rare among cryoEM structure cases. I don't think there is enough evidence to support this work can apply to "near-atomic resolution cryoEM structure" case which states in the abstract and discussion since the first version.

2. Concern to apply PRMT5 active site case to other structure

For ligand pose, water molecule often plays an important role by forming interaction network with active residues and the ligand. For PRMT5, luckily, no structural water is found in the active site for interaction so water molecule don't be considered in this reported approach. However, again, this is not a common case. Many ligands are surrounded by water molecules. For the feasibility of this computational analysis to other cases, water molecule should be addressed.

3. Concern of the validation approach to their docking result. As author claimed in the response letter, the computational approach finds a tighter inhibitor, 11-9F and show a 4-time better inhibitor in IC50. However, what their improvement is binding affinity but what they measure is IC50 which is based on a cell-based study. The cell-based inhibition can be influenced by other factors. That explains why 11-2F with a Kd of 82nM has only IC50 of 1 micromolar. Since author can directly determine Kd of 11-2F, the Kd of 11-9 in presence of MTA should be included as a key validation data for their docking result.

4. Unclearness of 2 angstrom shift, resulting in positive cooperativity. Even after two revisions, it is not unclear to me about this 2 angstrom shift. The shift is based on the alignment of TIM barrel. If author align the PRMT5-core domain, there won't be any shift. Since MTA and inhibitor are bound in PRMT5-core domain and PRMT5-core domain is where reaction occurs and MTA/inhibitor are bound, what is the rationale to align TIM-barrel and why this intra-domain movement leads to positive cooperativity. In addition, this movement can be caused by crystal packing since author align cryoEM structure with X-ray structure.

5. Concerns about docking pose to cryoEM map

Until now, it is still not clear to me why authors generate a selection of virtual ligand pose and fit into cryoEM data, while it can be done in the conventional way. The MTA density from cryoEM clearly showed the correct pose in Figure 3B. Ligand building simply based on the shape of density map is a common practice in protein crystallography and cryoEM. It is also common for both protein crystallography and cryoEM that sometimes different ligand poses can fit quite well in the

same density. Based on the local interaction, the one with more interactions is generally picked. This approach works well in protein crystallography and only requires some common senses of protein-ligand interaction. It is not clear to me the advantage of this computation analysis of ligand poses with cryoEM data-based selection compared to the common particle, which is cryoEM map analysis of ligand poses with interaction-based selection.

Minor issue

Can author provide any reference or evidence that ResMap tend to overestimate the resolution than the fold-standard FSC so they try not to report the active site local resolution.

Dear Editor,

We would like to thank reviewer #2 for his/her additional comments. We have addressed them as detailed in the next.

1. Concern to apply this “high resolution” example to other “low resolution” cases. The resolution of this work is 3.1 angstrom and for the active part is even better (~2.6-2.8 angstrom) because the active site is buried inside. First of all, the near-atomic resolution is >3.5 angstrom. Among 16694 cryoEM single-particle structures deposited in EMDB, only 2928 (less than 17.5%) structure has better than 3.1 angstrom resolution, 1218 (~7%) structures (has better than 2.8 angstrom resolution and 612 (3.6%) structures has better than 2.6 angstrom resolution. The PRMT5 case is a high resolution structure and, luckily, the resolution of active site is considered rare among cryoEM structure cases. I don't think there is enough evidence to support this work can apply to “near-atomic resolution cryoEM structure” case which states in the abstract and discussion since the first version.

Response: Thank you to the reviewer for classifying our structure in the high-resolution group. Yes, the resolution of the ligand allowed us to optimize the ligand orientation locally. On the other hand, the computational analysis was needed to distinguish the two poses that are flipped by 180 degrees as we presented in Supplementary Fig. 3E, which could not be done well by local refinement of the two poses. From the structural features, we suggested that even if the resolution is ~3.5 angstrom, the ligand poses may still be optimized within the binding pocket in a similar fashion when the cryo-densities allow the rough positions of the ligands into the binding pocket. We concur with the reviewer that this point needs to be further tested in other cases. We modified the abstract and the discussion to reflect this thinking (lines 37-38; 441-443).

2. Concern to apply PRMT5 active site case to other structure
For ligand pose, water molecule often plays an important role by forming interaction network with active residues and the ligand. For PRMT5, luckily, no structural water is found in the active site for interaction so water molecule don't be considered in this reported approach. However, again, this is not a common case. Many ligands are surrounded by water molecules. For the feasibility of this computational analysis to other cases, water molecule should be addressed.

Response: Yes, we are aware of the reviewer's concern, but at the resolution we have, it is impossible to model water molecules. Currently, in the resolution ranges of 3.5 to 2.0 angstroms, we are limited in quantifying the contributions of modeled water molecules to the binding site and the ligand fitting. On the other hand, the energy minimization in the computational analyses was based on commonly assigned partial charges that contained chemical consideration of ligand solubilization and would not have been further changed by the presence of water molecules. Instead, the water molecules were all stripped away at the beginning the computational analysis. The application of this procedure to other cases would still be possible. Further, when we consider the redesign of the compounds, we focused on chemical groups, instead of water molecules, because chemical groups may introduce more direct ligand-protein interactions and

will likely have stronger effects. Last, and probably more importantly, the ultimate test of the analysis will rest on whether the predicted modifications from computational analysis can really enhance potency of the ligands. The advantage of computational analysis is that it is fast and inexpensive and can incorporate many changes quickly. If better potency can be reached this way, it would really be worthwhile.

3. Concern of the validation approach to their docking result. As author claimed in the response letter, the computational approach finds a tighter inhibitor, 11-9F and show a 4-time better inhibitor in IC50. However, what their improvement is binding affinity but what they measure is IC50 which is based on a cell-based study. The cell-based inhibition can be influenced by other factors. That explains why 11-2F with a Kd of 82nM has only IC50 of 1 micromolar. Since author can directly determine Kd of 11-2F, the Kd of 11-9 in presence of MTA should be included as a key validation data for their docking result.

Response: Data for testing MTA-synergy of 11-9F in enzyme inhibition in cultured cells are now in Figure 7 (Figure 7F; lines 363-369). The PRMT5 activity was measured by quantifying symmetrical dimethylated arginine in a series of doses of 11-9F and in the presence of 0.0 and 5.0 micromolar MTA, respectively. After quantitative analysis with a single-binding site model, we found that the IC50 of 11-9F was increased by ~13x in the presence of 5.0 micromolar MTA when compared to that with no additional MTA. The synergy is a bit stronger than that for 11-2F (Fig. 1C) from SPR binding. We also note that before further tests, there is a need to enhance membrane permeability of 11-9F or develop a membrane-based delivery system to facilitate its crossing of plasma membranes, which is not part of the current work.

4. Unclearness of 2 angstrom shift, resulting in positive cooperativity. Even after two revisions, it is not unclear to me about this 2 angstrom shift. The shift is based on the alignment of TIM barrel. If author align the PRMT5-core domain, there won't be any shift. Since MTA and inhibitor are bound in PRMT5-core domain and PRMT5-core domain is where reaction occurs and MTA/inhibitor are bound, what is the rationale to align TIM-barrel and why this intra-domain movement leads to positive cooperativity. In addition, this movement can be caused by crystal packing since author align cryoEM structure with X-ray structure.

Response: For direct comparison, the alignment was first done at the bottom rim of the binding pocket of 11-2F in order to compare the shift in the upper part. The rationale is to look at the local differences when one end is aligned between the two structures.

If the crystal packing had an effect, it would have changed the global arrangement of all three domains. However, the global difference was found to be small when we did global alignment.

5. Concerns about docking pose to cryoEM map. Until now, it is still not clear to me why authors generate a selection of virtual ligand poses and fit into cryoEM data, while it can be done in a conventional way. The MTA density from cryoEM clearly showed the correct pose in Figure 3B. Ligand building simply based on the shape of density map is a common practice in protein crystallography and cryoEM. It is also common for both protein crystallography and cryoEM that sometimes different ligand poses can fit quite well in the same density. Based on the local

interaction, the one with more interactions is generally picked. This approach works well in protein crystallography and only requires some common senses of protein-ligand interaction. It is not clear to me the advantage of this computation analysis of ligand poses with cryoEM data-based selection compared to the common particle, which is cryoEM map analysis of ligand poses with interaction-based selection.

Response: We apologize that this point did not become clear to the reviewer when the two opposite poses were added to Supplementary Figure S3E in the last revision. At the current resolution, even though the active site resolution in ResMap is slightly better than the peripheral regions of the map, the ligand (11-2F) density contained two parts of similar shapes and dimensions, which made it difficult to distinguish the two opposite orientations of the ligand, as showed in Supplementary Fig. S3E. Real space refinement of the ligand against the density or the ligand/protein against the complex map could not distinguish the two opposite poses in the binding pocket. To help highlight this point better, we added a new panel at the bottom of the Supplementary Figure S3E to present these two poses in the binding pocket (after refinement). The binding pockets are the same even though the two poses are opposite to each other.

From such comparison and the different poses of 11-2F ranked by docking analysis, we learned that the computational analysis did make an important contribution for us to separate these two poses via energy minimization.

6. References on ResMap

Response: The comparison of resolution bins in ResMap in the panel 3 of figure 5 in the following reference showed that the ResMap may slightly overestimate local resolutions of the core regions. It is now in line 172 of the revised manuscript.

Jose Luis Vilas, Josué Gómez-Blanco, Pablo Conesa, Roberto Melero, José Miguel de la Rosa-Trevín, Joaquin Otón, Jesús Cuenca, Roberto Marabini, José María Carazo, Javier Vargas, Carlos Oscar S. Sorzano, MonoRes: Automatic and Accurate Estimation of Local Resolution for Electron Microscopy Maps, *Structure*, Volume 26, Issue 2, 2018, Pages 337-344.
[https://doi.org/10.1016/j.str.2017.12.018.\(https://www.sciencedirect.com/science/article/pii/S0969212617304434\)](https://doi.org/10.1016/j.str.2017.12.018.(https://www.sciencedirect.com/science/article/pii/S0969212617304434)).

7. Please provide references for line 113 that inhibitors have been "less potent in the presence of MTA" to emphasize the need for MTA cooperativity.

Response: There are multiple references stating that the current inhibitors are less potent in the presence of MTA, justifying their work for MTA-cooperative PRMT5 inhibitors. These are all very recent, highlight the timeliness of our work in this area. A few references are listed in the next. We added representative ones in the revised text in line 113.

- a. Smith, Christopher R., Ruth Aranda, Thomas P. Bobinski, David M. Briere, Aaron C. Burns, James G. Christensen, Jeffery Clarine et al. "Fragment-based discovery of MRTX1719, a synthetic lethal inhibitor of the PRMT5• MTA complex for the treatment of MTAP-deleted cancers." *Journal of Medicinal Chemistry* 65, no. 3 (2022): 1749-1766.
- b. Belmontes, Brian, Antonia Policheni, Siyuan Liu, Katherine Slemmons, Jodi Moriguchi, Hayley Ma, Daniel Aiello et al. "The discovery and preclinical characterization of the MTA

- cooperative PRMT5 inhibitor AM-9747." *Cancer Research* 82, no. 12_Supplement (2022): 1807-1807.
- c. Villalona-Calero, Miguel Angel, Amita Patnaik, Robert G. Maki, Bert O'Neil, James L. Abbruzzese, Ibiayi Dagogo-Jack, Siddhartha Devarakonda et al. "Design and rationale of a phase 1 dose-escalation study of AMG 193, a methylthioadenosine (MTA)-cooperative PRMT5 inhibitor, in patients with advanced methylthioadenosine phosphorylase (MTAP)-null solid tumors." (2022): TPS3167-TPS3167.
 - d. Levenets, Oleksandr, Anna Bartosik, Marta Sowińska, Karol Zuchowicz, Sujit Sasmal, Klara Korta-Piątek, Adam Radzimierski et al. "Discovery of novel MTA-cooperative PRMT5 inhibitors as targeted therapeutics for MTAP deleted cancers." *Cancer Research* 82, no. 12_Supplement (2022): 1806-1806.
 - e. Briggs, K., G. Corriea, A. Tsai, M. Zhang, M. R. Tonini, E. W. Wilker, C. B. Davis, K. M. Cottrell, J. P. Maxwell, and A. Huang. "24P Evidence for synergy between TNG908, an MTAPnull-selective PRMT5 inhibitor, and sotorasib in an MTAPnull/KRASG12C xenograft model." *Annals of Oncology* 33 (2022): S12.

Please let me know if more information is needed. Thanks.

Best regards,

Qiu-Xing